

# Examining the characteristics of aerosols: a statistical analysis based on a decade of lidar and photometer observations at the Eastern border of ACTRIS

Doina Nicolae[1], Gabriela-Ancuta Ciocan[1,2], Anca Nemuc[1], Victor Nicolae[1], Camelia Talianu[1], Jeni Vasilescu[1], Alexandru Dandocsi[1], Cristian Radu[1], Marius-Mihai Cazacu[3,4], Viorel Vulturescu[5], Livio Belegante[1]

[1] National Institute of Research and Development for Optoelectronics INOE2000, Magurele, 077125, Romania
[2] Faculty of Physics, University of Bucharest, Magurele, 077125, Romania
[3] Department of Physics, "Gheorghe Asachi" Technical University of Iasi, 700050 Iaşi, Romania
[4] INOESY SRL, 8 Fdc. Mestecanis Street, 707410, Valea Lupului, Iasi, Romania
[5] Theory of Mechanisms and Robots Department, Faculty of Industrial Engineering and Robotics, National University of Science and Technology POLITEHNICA Bucharest, Bucharest, Romania

*Correspondence to:* Anca Nemuc (anca@inoe.ro)

**Abstract.** A decade-long (2015–2024) analysis of aerosol properties was conducted at RADO-Bucharest station in Romania, a key atmospheric observational site at the Eastern border of the Aerosol, Clouds and Trace gases Research Infrastructure (ACTRIS). This study aims to characterize the optical and microphysical properties of aerosols, classify predominant aerosol types, and investigate their seasonal variability and transport pathways based on long-term multiwavelength Raman lidar and sun/sky/lunar photometer measurements.

Results indicate a dominance of fine-mode aerosols, with an average Aerosol Optical Depth (AOD) of ~0.2 and Ängström Exponent (AE) values between 1.5–2.0, highlighting pollution-driven aerosol regimes. Seasonal variations were observed, with continental aerosols prevailing in winter, dust transport peaking in spring (altitudes of 2–8 km), and biomass-burning aerosols increasing during summer. Marine aerosols were occasionally detected at ~2 km altitude, often mixed with dust. Analysis of 408 aerosol layers using the NATALI (Neural network Aerosol Typing Algorithm based on LIdar data) identified complex aerosol mixtures, with 63 high-resolution cases revealing a predominance of "dust polluted" and "continental smoke" types.

Lidar-derived extinction Ängström coefficients (median ~0.9 in the low troposphere) and lidar ratios (~48 sr in the low troposphere, ~49 sr in the high troposphere) suggest varying optical properties linked to aerosol composition and absorption characteristics.

FLEXPART (FLEXible PARTicle dispersion model) retro-plume simulations provided insights into aerosol source regions and transport patterns showing contributions from local emissions, long-range transported desert dust, and biomass burning events from Europe and North America.

These findings emphasize the persistent influence of regional pollution and transported aerosols on air quality and climate. The integration of ground-based remote sensing and advanced retrieval algorithms like NATALI provides a robust framework for aerosol characterization, enhancing climate models and air quality assessments.

## 1 Introduction

The Aerosol, Clouds, and Trace Gases Research Infrastructure (ACTRIS) coordinates activities to document, understand, and quantify the effects of short-lived atmospheric constituents on climate, air quality, human health, and ecosystems. ACTRIS consists of extensive atmospheric research facilities located across Europe and beyond. These facilities collect high-quality data on the distribution and variability of aerosols, clouds, and reactive trace gases in the atmosphere. ACTRIS helps to reduce



uncertainty in the life cycle of short-lived components and assess their potential consequences on climate, air quality, and ecosystems. It also examines how climate-ecosystem feedback loops may alter (Laj et al.,2024).

Aerosols are one of the key research issues of ACTRIS. A thorough understanding of aerosol particles is required to investigate the Earth's climate and temporal fluctuations. Tiny solid or liquid particles suspended in the atmosphere, known as aerosols,

represent a key element of atmospheric composition and can greatly influence both local air quality and broader environmental and climatic conditions  (Schwartz and Andreae, 1996;) Seinfeld 2016) Due to their relatively short atmospheric residence time—from several days to a few months—the processes of aerosol release and long-range movement are vital in the dynamics of the Earth's atmospheric system. Depending on their size and chemical makeup, aerosols can significantly alter the Earth's radiative energy balance by directly absorbing or scattering incoming solar energy. Moreover, they contribute indirectly to

climate dynamics by influencing cloud formation and behavior, as they often serve as nuclei around which cloud droplets or ice crystals can form(IPCC,2021);. Aerosol concentrations and features change greatly in the atmosphere due to their many origin sources and the short period they exist (Boucher, 2015). Scattering aerosols lowers the Earth's temperature by increasing its albedo, whereas absorbing aerosols raises the temperature budget and contributes to global warming (Moorthy et al., 2009; Sabetghadam et al., 2021). Aerosols are also capable of altering both the optical and the microphysical properties of clouds.

Numerous earlier investigations have examined how aerosols influence cloud development, rainfall, and radiation patterns by analysing their optical properties (Holben et al., 1998; Dubovik and King, 2000; Bréon et al., 2002).

The optical characteristics of aerosols might give valuable information for global climate change projections and response assessments (Hansen et al., 1997; Holben et al., 1998; Intergovernmental Panel On Climate Change, 2014). The optical depth of aerosols (AOD) is a key parameter used to quantify the degree to which aerosols in the atmosphere reduce sunlight by

scattering and absorption throughout a vertical air column (Bréon et al., 2002). Complementary to this, the Ångström exponent (AE) extinction related, which varies with wavelength, offers insights into aerosol particle size and classifications (Eck et.al, 1999; Gobbi et al., 2007) . Other crucial optical indicators include the single-scattering albedo (SSA), which describes the ratio of scattering to total extinction, and the asymmetry parameter or phase function, which defines the directional distribution of scattered light (Anon, 1999; García et al., 2016). The linear particle depolarization ratio from lidar observations is a key pa-

rameter for aerosol typing, distinguishing between spherical and non-spherical particles. It enables the identification of different aerosol types, such as dust, smoke, and volcanic ash, and can differentiate between fresh and aged aerosols, improving atmospheric monitoring and analysis.  The lidar ratio, defined as the ratio between the aerosol extinction coefficient and the backscatter coefficient, is an important parameter in characterizing aerosol properties. It provides insights into the scattering and absorption behavior of particles in the atmosphere. High lidar ratios typically indicate aerosols that strongly absorb light,

such as smoke or urban pollution, while lower lidar ratios are associated with more scattering aerosols like dust or marine particles. By analyzing the lidar ratio, additional information on the composition, size, and optical behavior of aerosol can be retrieve, contributing to improved aerosol classification and environmental assessments.

Technologies based on remote sensing, including both satellite observations and instruments on the ground, have significantly advanced the study of global climate. These tools enable detailed analysis for climate simulations, monitoring changes in land

surfaces, and examining how human-induced pollution impacts ecosystems, aquatic environments, and atmospheric conditions. (Dubovik et al., 2021; Yang et al., 2013). Satellite-based observations enable broad-scale monitoring of aerosol properties across the globe. While they deliver long-term and wide-ranging datasets, the precision of the information obtained is limited by the spatial resolution of the sensors and the algorithms used for data retrieval. (Kaufman et al., 2002; Dong et al., 2023). While ground-based remote sensing systems cover a smaller geographical area, they tend to produce more detailed and

accurate measurements. (Zhao et al., 2005; Ali et al., 2014; Cimini et al., 2020). Aerosol radiation, climatic impacts, and air quality are all substantially influenced by the vertical distribution of aerosols, as well as the presence of lofted layers above the boundary layer, which are frequently created by long-range transport and layer mixing. Passive remote sensing from space cannot resolve these vertically connected characteristics or aerosol composition. Active sensors, such as lidars, can regularly



and constantly characterize the vertical structure of smoke plumes (Engelmann et al., 2016; Baars et al., 2016). The launch of
the CALIPSO (Cloud-Aerosol Lidar and Infrared Pathfinder Satellite Observation) in 2006 (Winker et al., 2009; Omar et al.,
2009) made it feasible to examine the vertical distribution of aerosols on a worldwide scale. However, its low return rate and
limited geographical coverage prevent daily 3D aerosol climatology analysis on a global scale. Several new satellite missions
have the objective to retrieve products on profiles for global observations, as is the case of Aeolus for wind profiles from the
surface up to 30 km altitude. An extensive increase of available information related to aerosols and clouds distribution is
expected for the recent EarthCARE (Earth Cloud Aerosol and Radiation Explorer) mission, focused on aerosol and cloud
optical property profiles, as well as their classification. ATLID is the lidar onboard, with a three-channel, linearly polarized,
high-spectral-resolution capabilities in ultraviolet (Donovan et al., 2024).

Lidars measuring fromthe ground level serve as a valuable counterpart to satellite-based systems by enabling continuous mon-
itoring of the boundary layer's daily patterns and diurnal variations, along with vertical transport and mixing phenomena. Data
collected from ground-based lidar systems at a single location can reflect aerosol behavior in the free troposphere over a broad
regional scale—typically ranging from 100 to 300 kilometers—as evidenced by findings from the European Aerosol Research
Lidar Network (EARLINET) and corroborated by CALIPSO satellite lidar data. (Pappalardo et al., 2010). EARLINET (Pap-
palardo et al., 2014) has been providing aerosol optical property profiles at the continental scale since 2000. In ACTRIS, the
synergy between aerosol profile and column observations is exploited. The aerosol high-power aerosol lidars are used to pro-
vide profile aerosol optical properties (aerosol backscatter coefficient, aerosol extinction coefficient, and aerosol linear depo-
larization ratio) at one or more wavelengths, allowing the subsequent calculation of several spectral parameters (Ångström
exponents, lidar ratios) of the lofted aerosol layers, and thus aerosol classification. Automatic sun/sky/lunar photometers and
retrieval techniques provide column aerosol properties both directly (e.g., daytime and nighttime spectral extinction AOD and
daytime downward sky angular, spectral, and polarized radiance) and indirectly (size distribution, refractive indexes, single
scattering albedo, spherical fraction, scattering properties) (Dubovik and King, 2000; Dubovik et al., 2014).

The Centre for Aerosol Remote Sensing (CARS) is responsible to guide the Quality Assurance (QA) of both lidar and pho-
tometer observations. The ACTRIS-CARS component employs rigorous protocols to ensure the accuracy and reliability of
aerosol measurements, including standardized calibration procedures, regular intercomparisons with reference systems, and
periodic analysis of QA tests to evaluate the performance of each lidar instrument. Among the tests used for routine checkups
are the telecover test (to assess near-range alignment), the Rayleigh Fit test (for far-range alignment), polarization calibration
tests (to evaluate systematic errors in the polarization channels), the dark test (to check data acquisition performance), and the
range bin test (to correct range offset). Recently, quality-controlled 20-year climatological datasets were released.

The rigorously validated datasets provided by EARLINET-ACTRIS have played a key role in numerous research areas. These
include the classification of aerosol types and analysis of their optical characteristics(Nicolae et al., 2018; Papagiannopoulos
et al., 2018), retrieval of microphysical parameters using inversion techniques (Müller et al., 2016), enhancement of optical
information by combining lidar and sun photometer measurements(Chaikovsky et al., 2016), investigation of aerosol water
uptake behavior(Bedoya-Velásquez et al., 2018), and of interactions between aerosols and clouds (Mamouri and Ansmann,
2017; Marinou et al., 2019). Additionally, these datasets have contributed to improving the accuracy of satellite data prod-
ucts(Amiridis et al., 2013; Floutsi et al., 2023), have been used to refine radiative transfer modeling(Granados-Muñoz et al.,
120   2019).

Europe often serves as a convergence zone for pollutants carried over long distances, characterized by a diverse composition
that includes both human-made and naturally occurring particles. The vertical structure of aerosols in this region is frequently
layered, with varying degrees of mixing between the surface and elevated atmospheric levels. Common aerosol types observed
over Europe encompass mineral dust originating from North Africa and the Middle East, industrial emissions from central
European regions, locally generated pollutants, continental background aerosols, and smoke from biomass burning, sourced
both nearby and from remote locations. (Nicolae et al., 2019). In 2010, (Pappalardo et al., 2013) conducted coordinated



measurements to give insight on the dynamics driving the transit of Icelandic ash plumes following the eruption of the Ey-jafjallajökull volcano. Instrumental deployments at EARLINET-ACTRIS have been employed in various recent volcanic erup-tions to characterize volcanic particle characteristics (Lieke et al., 2013) or novel particle production in volcanic plumes (Rose et al., 2019). (Amiridis et al., 2023) conducted a demonstration study that leveraged synergies between space and ground-based remote sensing to estimate volcanic ash movement and hence aircraft safety. Furthermore, EARLINET-ACTRIS con-tributes to desert dust atmospheric composition research as part of the WMO's Sand and Dust Storm Warning Advisory System (SDS-WAS) for the evaluation and optimization of desert dust predictions (Binietoglou et al., 2015).

Furthermore, the EARLINET-ACTRIS network plays a key role in observing and documenting thick smoke layers originating from large-scale wildfires in regions such as Canada, Australia, Siberia, and the Mediterranean. These observations are essen-tial for precisely evaluating how smoke is transported through the atmosphere and its impact on radiative processes in both the troposphere and the stratosphere.

In reality, it is difficult to accurately identify and distinguish between these various sources. (Kaufman et al., 2001, 2005; Pöschl, 2005; Dubovik et al., 2019). Various methodologies, approaches, and computational procedures can be employed to acquire the optical and microphysical properties of aerosols. Typically, these algorithms employ distinct datasets. Certain algorithms utilize data obtained from a single instrument. In this scenario, the only requirement is the ability to operate at many wavelengths, ensuring that the available physical data are adequate for the retrieval process.

The AErosol RObotic NETwork (AERONET) inversion code uses only sun/sky-photometer data (Dubovik and King, 2000). Similarly, the Raman technique and regularization algorithm use only LIDAR data (Ansmann et al., 1990; Weitkamp, 2005; Veselovskii et al., 2002). The POlarization LIdar PHOtometer Networking (POLIPHON) relies on simultaneous measurement of the lidar measured aerosol backscatter and linear particle depolarison ratio to discriminate between various aerosol types (Tesche et al., 2009; Mamouri and Ansmann, 2014; Sicard et al., 2016). The Neural Network Aerosol Typing Algorithm based on Lidar Data (NATALI) uses the ability of specialized ANN (Artificial Neural Networks) to resolve overlapping values of intensive optical parameters obtained for each recognized layer in multiwavelength Raman lidar profiles: three backscatter coefficients ($\beta$), two extinctions ($\alpha$), and one linear particle depolarization ratio ($\delta$). A similar method was developed by (Papagiannopoulos et al., 2018), but making use of the Mahalanobis distance function instead of neural networks.

However, all the above retrieval techniques have limitations (Mylonaki et al., 2021) In case of AERONET typing, data are limited to daytime observations and layers of different types cannot be resolved. In case of POLIPHON, hard assumptions must be applied. In case of NATALI, data are limited to nighttime observations. Also, these algorithms can retrieve the aerosol type but not the microphysical properties of aerosols in a quantitative way. An approach that proved to be useful in resolving aerosol mixtures and layers was the use in synergy of active and passive remote sensors. Approaches that utilize data from multiple instruments integrate observations from various sources to extract detailed aerosol microphysical characteristics. These techniques require the spatial and temporal alignment of measurements from the different devices to ensure consistency and accuracy.The LIdar-Radiometer Inversion Code (LIRIC) extracts aerosol optical and microphysical properties at different heights for fine and coarse modes (Wagner et al., 2013; Granados-Muñoz et al., 2014; Chaikovsky et al., 2016). The algorithm utilizes AERONET inversion results, including column volume concentration, volume specific backscatter, and extinction coefficients, as a priori knowledge (Chaikovsky et al., 2016).The products include of backscatter and volume concentration profiles, Ängström exponent (AE) values, as well as LIDAR (LR) and depolarization ($\delta$) ratios. The Generalized Aerosol Retrieval from Radiometer and LIDAR Combined data (GARRLiC) achieves a more profound synergy between the LIDAR and sun/sky-photometer data. GARRLiC utilizes the technique developed by (Lopatin et al., 2013) to simultaneously invert the coincident LIDAR and sun/sky-photometer radiometric data. Another notable difference between GARRLiC and LIRIC is the reversal of two separate aerosol modes, enabling the independent retrieval of aerosol optical and microphysical param-eters for both the fine and coarse modes (Córdoba-Jabonero et al., 2018). GARRLiC has been integrated as a component of



Generalized Retrieval of Aerosol and Surface parameters (GRASP), a comprehensive method used to analyze atmospheric parameters obtained from various remote sensing observations (Dubovik et al., 2011; Dubovik and King, 2000).

The establishment of the ACTRIS ERIC opens in Europe new prospects for long-term synergistic studies on aerosol properties (Laj et al., 2024). One particular requirement for the ACTRIS aerosol remote sensing observational platforms refers to the colocation of aerosol high-power lidars and the sun/sky/lunar photometers, which makes possible the retrieval of aerosol microphysical properties at a much larger time and spatial scale than before. Some of the ACTRIS observatories are operational for many years, however only recently the operational and quality assurance procedures have been harmonised and implemented at continental scale, ensuring the full quality control and traceability of the data products which enable the implementation of GARRLIC/GRASP retrievals at all sites. RADO-Bucharest facility is one of the few ACTRIS aerosol remote sensing sites with historical collocated lidar and photometer measurements, and the only one in East Europe. With this study we want to emphasize the added value of coupling long-term ground-based remote sensing observations with advanced retrieval algorithms and model simulations to resolve aerosol properties in complex mixtures and environments.

## 2        Data and methodology

This paper presents the characteristics of the aerosol layers measured in South-East Romania by means of collocated sun/sky/lunar photometer and multiwavelength Raman lidar during a 10 years period, from 2015 to 2024. Optical and microphysical properties as well as predominant aerosol types are retrieved using single instrument algorithms and multi-instrument methodologies, and analysed against FLEXPART (FLEXible PARTicle dispersion model) simulations.

RADO-Bucharest, where the measurements were performed is one of the four ACTRIS observational platforms established in Romania, hosting and operating scientific instruments and laboratories for atmospheric research. It is located 8 km southwest of Bucharest (44.340N; 26.010E, 73m a.s.l.) in a flat terrain, surrounded by agricultural fields and residential areas (Nicolae et al., 2010). It is a regional WMO-GAW station featuring aerosol in situ, aerosol remote sensing and cloud remote sensing equipment, complemented by trace gases and weather sensors (Carstea et al., 2019; Pîrloagă et al., 2022). The research performed here focuses on observing, studying, and understanding the changes and interactions between climate-relevant atmospheric variables and components. The Aerosol Remote Sensing Laboratory is an ACTRIS-compliant facility comprising aerosol high-power lidars of various designs (multi-wavelength, Raman, scanning, high spectral resolution) and a co-located sun/sky/lunar photometer. Research focuses on characterising the optical and physical properties of the aerosol layers within the atmospheric column. Data are submitted in near real time to ACTRIS Data Centre (EARLINET database for the lidars, and AERONET database for the photometer), being further used for model validation (Stebel et al., 2021), satellite Cal/Val (Proestakis et al., 2019) and scientific studies (Nicolae et al., 2018; Tsekeri et al., 2023).

The multiwavelength polarization Raman lidar instrument (RALI), located at the INOE site, is designed for high-resolution atmospheric measurements. It utilizes Raman scattering to measure aerosol properties such as extinction profiles, backscatter coefficients, and water vapor content. This lidar operates at multiple wavelengths, typically 355 nm, 532 nm, and 1064 nm, and is equipped with polarization channels for aerosol typing. The instrument provides detailed aerosol profiles from 600 m to 20 km in altitude. It features a raw temporal resolution of 10 seconds and a vertical resolution of 3.75 meters. RALI can operate during daytime using elastic and polarization channels, while additional extinction information is retrieved from Raman channels (355 nm and 532 nm) during nighttime. The processing is done using the Single Calculus Chain (SCC), which outputs aerosol optical parameters with a 30-minute temporal average and a 15-meter vertical resolution (Nicolae et al., 2018).

The sun/sky/lunar photometer, used to measure spectral solar and lunar irradiance, as well as sky radiances, consists of a sensor head equipped with 25 cm collimators mounted on a 40 cm robotic base. The system is programmed to automatically track the Sun, sky, and Moon according to a pre-set routine. The instrument is supported by a weatherproof plastic case containing a control unit, batteries, and satellite transmission. It operates across spectral bands from 340 to 1640 nm, with a Si detector covering the range from 340 to 1020 nm, and an InGaAs detector covering 1020 to 1640 nm. Direct sun measurements are



performed in eight spectral bands within approximately 10 seconds, using interference filters at specific wavelengths, including a 940 nm channel for determining column water vapor. The measurements begin at 7 in the morning and continue until evening, with the optical depth calculated through spectral extinction using the Beer-Bouguer Law. (Holben et al., 1998) provides a full instrument description of the first standard AERONET photometer.

Three direct sun measurements are taken consecutively at 30-second intervals, producing a "triplet" observation for each wavelength. For larger air masses, measurements are taken at 0.25 air mass intervals, while at smaller air masses, they are spaced 15 minutes apart to minimize cloud interference. Additionally, sky radiance is measured in four spectral bands (440, 670, 870, and 1020 nm) along both the solar principal plane and solar almucantar. This provides data on aerosol optical depth, size distribution, and phase function. Up to nine solar principal plane and six almucantar measurements are performed daily, with

the latter being used to retrieve aerosol properties for particles in the size range of 0.1 to 5 µm (Dubovik and King, 2000).

In this study, we used the NATALI algorithm on multiwavelength Raman lidar data collected with the RALI system to capture the typical properties of aerosol layers in the lower and higher atmosphere. The aerosol model used to train the neural networks combines the Global Aerosol Data Set (Köpke et al., 1997) with the T-matrix numerical approach (Mishchenko et al., 1996; Waterman, 1971) to iteratively compute the intensive optical characteristics of each aerosol type: extinction related Ängström

coefficient (AE) at 355 to 532 nm, backscatter related Ängström coefficients - also known as color indexes (CI) at 355 to 532 nm and 532 to 1064 nm respectively, and lidar ratios (LR) at 532 and 355 nm. Each distinct aerosol category is modeled as a homogeneous blend of basic components that do not engage in any physical or chemical interactions, and the proportions of these components can vary (through mixing ratio). These basic components are obtained from the OPAC database and include water-soluble particles, insoluble matter, soot, different sizes of mineral dust (nucleation, accumulation, coarse), sulfates, and

sea salt (accumulation and coarse modes).To replicate aerosol anisotropy, particles were modeled as spheroids with varying aspect ratios (the ratio of polar to equatorial lengths). The NATALI model utilizes standard parameters typically available in the EARLINET dataset, such as backscatter coefficient ($\beta$) profiles at wavelengths of 1064, 532, and 355 nm; extinction coefficient ($\alpha$) profiles at 532 and 355 nm; and, when available, linear particle depolarization ratio ($\delta$) profiles at 532 nm. Determining the specific aerosol type can be complex due to variables such as the presence or absence of $\delta$ data and the quality

of optical measurements, which are influenced by factors like calibration and inherent measurement uncertainties.. To address these complexities, two separate classification methods are applied—each offering a different level of resolution for aerosol identification. When depolarization data is included and all optical parameters meet strict quality criteria (e.g., extinction coefficient uncertainty ≤50%, backscatter coefficient uncertainty ≤20%, depolarization ratio uncertainty ≤30%), a high-resolution classification is possible. This allows mixtures to be resolved into 14 potential outcomes, including pure types (≥90%

purity), binary mixtures, and ternary mixtures. In contrast, when depolarization data is not available, the classification is performed with lower resolution, meaning that mixed aerosol types cannot be distinguished.. The main aerosol category is determined across six possible outputs, each representing a dominant type that may include up to 50% presence of other aerosol components as minor traces: continental, continental polluted smoke, dust, marine, and mineral mixtures. The correspondence of the aerosol types provided by NATALI in high- and low-resolution modes is presented in Figure 7 (left panel).

It is important to highlight that the aerosol types recovered by NATALI still may be accompanied with small proportions of other types (less than 10%), even in high resolution mode. In low resolution mode, the prevalence of a specific type is considered when that type is recognized by the neural networks in over 50% of the combinations of aerosol intensive parameters within the layer. This means that if, for instance, continental and smoke are mixed together, NATALI will identify it as continental if smoke is not the dominating component, or as smoke if smoke is the major component. Furthermore, the spectral

datasets obtained from the lidar often lack sufficient physical information content, along with their accompanying uncertainty, to accurately differentiate between fundamental aerosol types such as continental pollution and smoke, or marine and mineral mixtures. Such ambiguous cases are excluded from our analysis, based on the NATALI's built-in flagging, which identifies and highlights uncertain retrievals in the comments sections of the output file.



NATALI has been used in European studies (Nicolae et al., 2019) on smaller areas with a few lidar station data (Voudouri et
al., 2019; Talianu and Seibert, 2019; Mylonaki et al., 2021), and in comparisons with other automated typing algorithms
(Veselovskii et al., 2020; Srivastava et al., 2021). Furthermore, NATALI's thresholds for lidar optical derived parameters or
as a method for data typing have been frequently used, even beyond multiwavelength Raman lidar data (Li et al., 2022; Szczep-
anik et al., 2021; Wang et al., 2019; Siomos et al., 2020).

In this study we have used 218 multiwavelength Raman lidar datasets measured between 2015 and 2024. Optical profiles were
computed with the EARLINET's Single Calculus Chain (D'Amico et al., 2016; Mattis et al., 2016) and used as input to the
NATALI software. A number of 408 aerosol layers have qualified for the retrieval, out of which 229 were typed without any
flag regarding the too high uncertainty of at least one of the intensive parameters, or low confidence of the typing made by the
neural networks. Out of these, 63 presented calibrated linear particle depolarisation ratio values, and therefore allowed the
high-resolution mode typing.

Column aerosol properties and types are retrieved from the sun/sky/lunar photometer using the AERONET algorithms. Aerosol
Optical Depth (AOD) data were collected at three quality levels: Level 1.0 (unscreened), Level 1.5 (cloud-screened), and Level
2.0 (cloud-screened and quality-assured). The automatic cloud-screening algorithms, developed by (Smirnov et al., 2000) and
(Alexandrov et al., 2004), rely either on the lower temporal variability of AOD compared to clouds or on the nearly wave-
length-independent optical depth of clouds.

In this study, we used AERONET Version 3 retrievals at Level 2.0 for both direct and inverse products to reduce potential
uncertainties. In addition to data gaps caused by instrument calibration, interruptions in the time series may also occur due to
cloudy conditions or instrument malfunctions.

AERONET data are commonly used for aerosol classification through cluster analysis, which explores relationships between
two or multiple parameters. Building on previous work by (D'Almeida et al., 1991), (Dubovik et al., 2002), and (Toledano et
al., 2007), we used AOD at 440 nm and the Ängström Exponent (calculated between 440 and 870 nm) to define threshold
values for aerosol clusters.

However, the inability of this method to distinguish between absorbing and non-absorbing anthropogenic aerosols without
geographic context led to improvements. In 2010, (Lee et al., 2010) introduced additional thresholds, emphasizing the rela-
tionship between Single Scattering Albedo (SSA) at 440 nm and Fine Mode Fraction (FMF) at 550 nm. FMF provided a more
quantitative measure of particle size compared to the Ängström Exponent, which gives more qualitative information.

Further refinements by (Zhang and Huo, 2016) and (Logothetis et al., 2020) incorporated the Ängström Exponent along with
two new threshold values to better classify mixed aerosols.

FLEXPART - FLEXible PARTicle dispersion model is a Lagrangian particle dispersion model designed to simulate the motion
of particles in space, the direction of particles' movement and the transformation of particles' properties under various envi-
ronmental conditions. By simulating the diffusion processes of aerosols in the atmosphere, FLEXPART can help to identify
the footprint of aerosol particles, such as the sources and the dispersion paths. In this study, to classify the aerosol source
regions, a statistical analysis of FLEXPART "retroplumes" (back-trajectories and emission sensitivity) was performed. The
retroplumes were obtained from FLEXPART model version 10.4 (Pisso et al., 2019) coupled with ERA5 meteorological data
provided by the European Centre for Medium-Range Weather (ECMWF). FLEXPART was run in the backward mode for a
simulation period of 240 hours (10 days) starting at every 3 hours, with an emitted particles history interval of 30 minutes. The
first version of the Lagrangian particle dispersion model FLEXPART, developed in the mid-1990s, was meant to calculate the
long-range and mesoscale dispersion of hazardous compounds from point sources, such as those discharged following a nuclear
power plant accident (Stohl et al., 1998, 2005). Its applications have expanded to include a wide spectrum of atmospheric
gases and aerosols, including greenhouse gases, short-lived climate forcers such as black carbon and volcanic ash, as well as
the atmospheric branch of the water cycle. Given appropriate meteorological input data, it can be employed on scales ranging
from dozens of meters to global. In particular, FLEXPART's inverse modeling based on source-receptor connections is



commonly used. FLEXPART 10.4 improves performance, physicochemical parameterizations, input/output formats, and pre-processing and post-processing software. This version uses ECMWF Integrated Forecast System meteorological input data and data from the United States National Centers of Environmental Prediction (NCEP) Global Forecast System (GFS). To

account for the vertical velocity distribution asymmetry (upward and downward air movements) and air density gradients in the convective boundary layer, a unique turbulence model was developed. The aerosol wet deposition methodology has been completely revised, and a more complex gravitational settling parameterization has been added. Backward simulation of atmospheric concentrations at receptor locations with FLEXPART helps understand lidar and photometer aerosol compositions at a site (Pisso et al., 2019).

## 3      Results and discussions

### 3.1      Sun-sky/lunar photometer retrievals

The radial time series visualizations (Fig. 1) reveal distinctive characteristics of aerosol properties over multiple years. Across all three metrics—AE, AOD, and FMF—there appears to be minimal variation in seasonal patterns, suggesting a consistent aerosol behavior throughout the observed period. The AOD values (Fig. 1 left panel) are notably low, indicating a relatively

small concentration of aerosols in the atmosphere at this location. In contrast, the high AE (Fig. 1 center panel) and FMF (Fig. 1 right panel) values suggest a predominance of fine particles, likely indicative of pollution sources or fine-mode aerosol components rather than larger dust particles. These trends imply a stable, fine-mode dominated aerosol regime, with limited influence from coarse-mode aerosols or seasonal fluctuations.

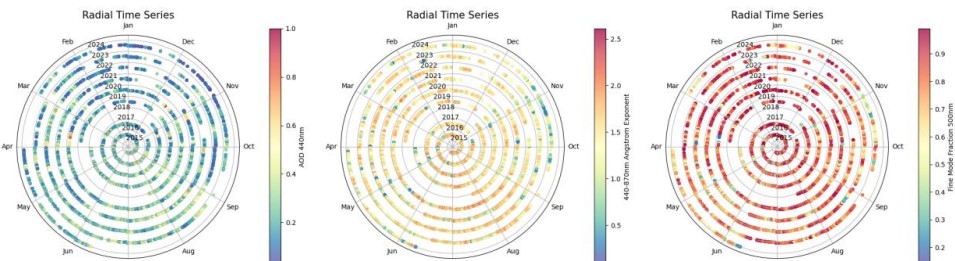

**Figure 1: Radial time series visualizations of Level 2 aerosol optical depth (AOD) at 440 nm (left panel), Ängström Exponent (AE) values calculated between 440-870 nm (center panel), and Fine Mode Fraction (FMF) at 500 nm (right panel) from 2015 to 2024**

Following the radial time series analysis, heatmaps (Fig. 2) were generated to further explore the seasonal and yearly variations in AOD, AE, and FMF. The AOD heatmap (Fig. 2 upper left panel) shows slightly higher mean values during the summer months, while winter exhibits lower AOD values, consistent with an overall low AOD range. This seasonal trend in AOD

aligns with typical atmospheric conditions, where warmer seasons often see increased aerosol presence due to natural and anthropogenic activities. In contrast, the AE heatmap (Fig. 2 upper right panel) highlights consistently high values throughout the years. Two exceptions occur in April 2018, February 2021, September and October 2023. This might be due to the number of datasets which is considerably lower, therefore enabling a bias to the measurement period, the values not being able to stabilize to the common behavior. Meanwhile, the FMF heatmap (Fig. 2 lower left panel) reveals minimal seasonal or

interannual variation, reinforcing the observation that fine-mode aerosols remain stable throughout the study period. Additionally, an extra heatmap was created to display the percentage of times AE values exceeded 1 (Fig. 2 lower right panel), further illustrating the persistent dominance of fine particles across the timeline.



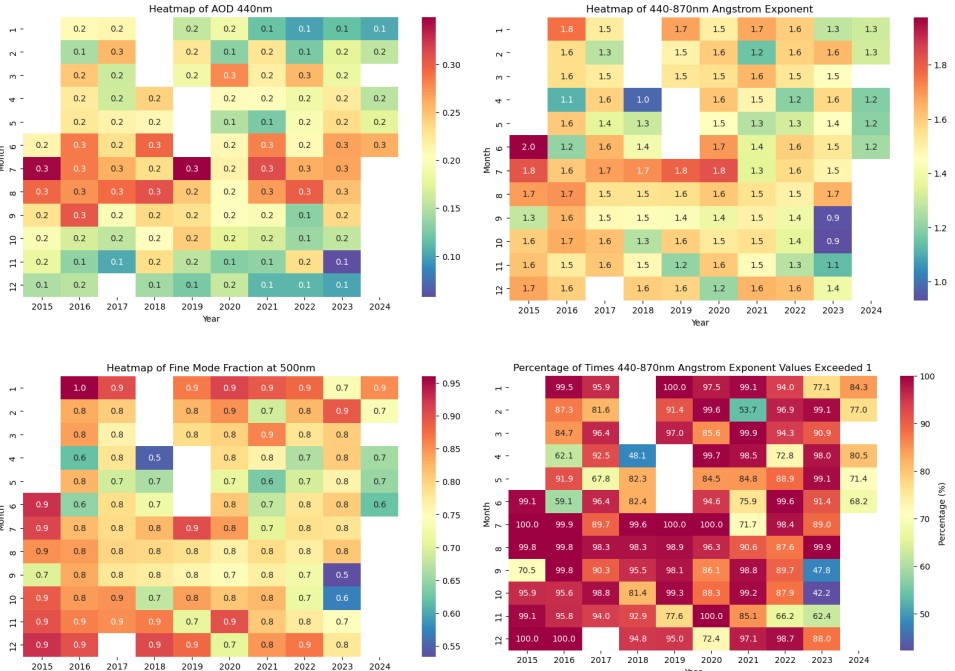

**Figure 2: Heatmap visualization of monthly mean Level 2 values: AOD at 440 nm (upper left panel), Ängström Exponent values calculated between 440-870 nm (upper right panel), Fine Mode Fraction at 500 nm (lower left panel), and the monthly percentage of occurrences when the 440-870 nm Ängström Exponent exceeded 1 (lower right panel).**

The aerosol classification analysis (Fig. 3) revealed distinct clusters that characterize the aerosol types at the study location. As it can be seen in the left panel of Fig. 3, most of the data points clustered around an AOD of 0.2 and an AE range of 1.5–2.0, indicating a dominant presence of fine-mode aerosols, typical of polluted or continental aerosol regimes. The main clusters identified were the Tight Continental, Polluted, and Mixed clusters, each representing different aerosol characteristics in terms of size and composition. In contrast, a more disperse Dust cluster was observed, which likely represents the sporadic presence of larger, coarse-mode particles.

Notably, particularly strong dust events were recorded in 2016, 2018, and 2021, which correspond to periods of increased dust transport or local dust activity, this can be clearly seen in Fig. 3 right lower panel. The Continental cluster, on the other hand, showed a steady increase over the years, suggesting a gradual shift toward a more continental aerosol regime. This increase may be attributed to changing atmospheric conditions or shifts in regional pollution sources.

A significant drop in the Continental aerosol type was observed in 2018 (Fig. 3 right upper panel), which can be explained by the higher number of datasets collected during the summer months. During this time, the aerosol type distributions were more complex, with an increased presence of mixed and polluted aerosol types, likely due to enhanced anthropogenic activity and atmospheric conditions favorable for the formation of mixed aerosol populations.





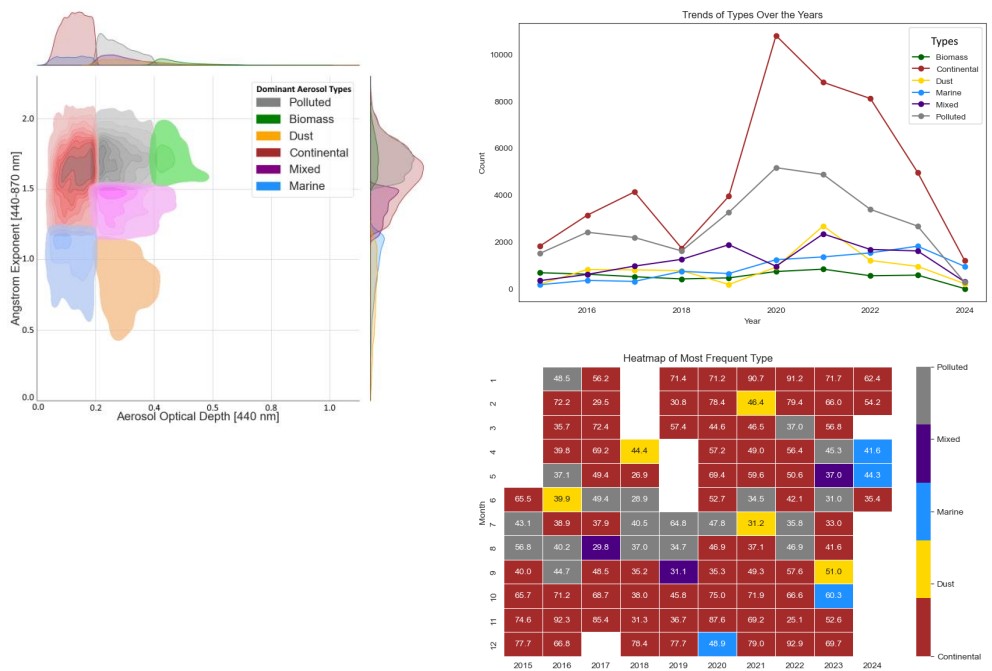

**Figure 3: Aerosol classification analysis based on three visualizations. The left panel presents the classical distribution plot of aerosol types. On the right, the upper panel displays the yearly mean values for each aerosol type, while the lower panel shows a monthly heatmap of the percentage occurrence of each aerosol type.**

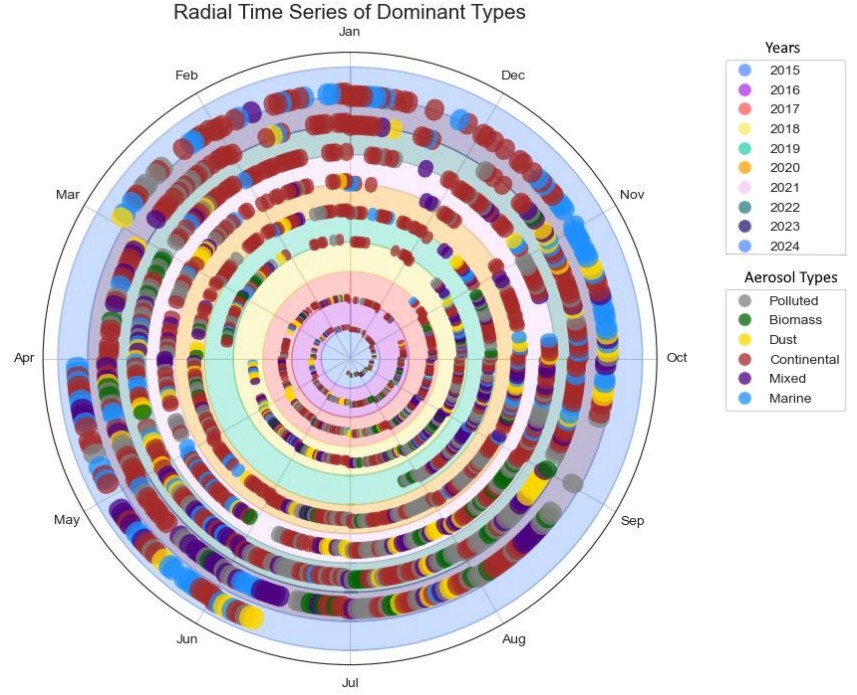

**Figure 4: Radial time series of daily predominant aerosol types. A specific aerosol type is considered predominant if it exceeds half of the daily data points.**



The radial time series of aerosol classification, presented in Fig. 4, also revealed temporal patterns in the occurrence of different aerosol types throughout the study period. Dust events, which are typically associated with larger, coarse-mode particles, were observed in several instances, with particularly long dust events recorded in 2017, twice in 2021, and again in 2022, as it can be seen in Fig. 4. However, these longer dust events were not long enough to count as the dominant aerosol types of their respective months. Notably, higher concentrations of dust were observed in the spring months, aligning with seasonal dust transport patterns.

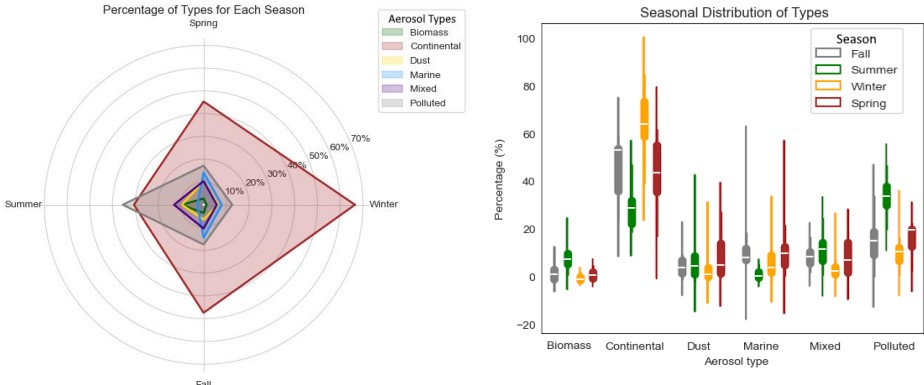

**Figure 5: Radial visualization of seasonal percentages of aerosol types (left panel), box-plot seasonal distribution of types (right panel)**

In 2021, a particularly long Marine event was recorded, indicative of air masses originating from oceanic regions, bringing a distinct aerosol composition. Marine aerosols were also present but in a split distribution during the spring and fall, suggesting that seasonal changes in atmospheric conditions influence the proportion of marine aerosols, with more complex dynamics in these transitional seasons, this can be better observed in the left panel of Fig. 5.

Biomass burning events were noted periodically, particularly in July and August, corresponding with typical biomass burning seasons in many regions. These events were also characterized by higher concentrations of Polluted, and Mixed aerosol types, especially during the summer months, when elevated temperatures and human activities lead to increased emissions. This seasonality in biomass burning events is aligned with the enhanced aerosol concentrations typically observed in summer due to anthropogenic activities and natural fires.

The concentration of Continental aerosols was higher during the winter months, suggesting a shift in the aerosol regime toward more stable, fine-mode particles in colder seasons. This could be related to the dominance of regional pollution sources or atmospheric conditions favoring the persistence of continental aerosols during the winter. As it can be seen in the right panel of Fig. 5, the Continental type stands out, with a significantly higher occurrence, particularly in fall and winter, compared to other aerosol types. In contrast, types like Marine, Mixed, and Dust exhibit a more balanced distribution across seasons, with lower and more consistent percentages. The seasonal variation for Polluted aerosols is also notable, particularly with a peak in summer.

### 3.2 Multiwavelength lidar retrievals

The optical data products obtained from the multiwavelength Raman lidar, including backscatter coefficient profiles at 1064, 532, and 355 nm, extinction coefficient profiles at 532 and 355 nm, and linear particle depolarisation ratio profile at 532 nm, were processed using the Single Calculus Chain. Level 2 processed data (which have passed all the quality assurance criteria) were then downloaded from the EARLINET-ACTRIS data portal and used as input for NATALI (Nicolae et al, 2024). The aerosol layers were classified as either low or high troposphere based on the altitude of the layer boundaries. In the following,



"low troposphere" (LT) means the first identified layer closest to the ground, having the bottom above the minimum product height but below the climatological height of the Planetary Boundary Layer (PBL). Minimum product height represents the altitude above which the optical products (backscatter and extinction coefficients) are all valid, i.e. above the full overlap of the instrument (maximum for all channels), so that the intensive parameters can be quantified. The minimum product height for the placement of RADO-Bucharest is 500 meters, while the maximum height for the first layer bottom (calculated as the climatological PBL height) is 1300 meters. Furthermore, a stringent limitation has been imposed on the upper boundary of the layers in the lower troposphere, requiring them to be situated no higher than 3000 m above sea level. This implies that thick layers, even if their lower boundary is below the typical height of the planetary boundary layer, are not taken into account in the lower troposphere. The term "high troposphere" refers to all the layers that are located above the first layer. Particles created locally are typically carried by layers in the low troposphere. Layers situated in the upper troposphere are regarded as fully disconnected from the Earth's surface and have the potential to contain particles that have been transported across long distances. For each layer identified in the low and in the high troposphere, we have calculated the layer-mean aerosol intensive parameters, i.e. extinction (AE) and backscatter (CI) related Ängström coefficients, lidar ratios (LR) and linear particle depolarisation ratio (DEP).

Figure 6 presents the distribution, central tendency, and spread of the aerosol extinction related Ängström coefficient at 355 to 532 nm, and of the lidar ratio at 355 nm.

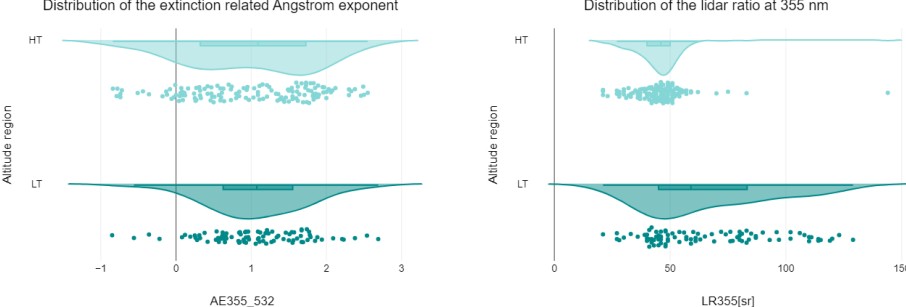

**Figure 6: Distribution, central tendency, and spread of the aerosol optical properties: a) extinction related Ängström coefficient (left panel); b) lidar ratio at 355 nm (right panel)**

We observe that in the low troposphere (LT) the extinction related Ängström coefficient has a mono-modal aspect of the distribution and a median around 0.9. It means that particles closer to the ground are medium-sized, and their dimension is almost uniformly spread around the median, although slightly skewed towards larger particles. In the high troposphere (HT), the distribution of the extinction related Ängström coefficient has a bi-modal aspect, with medians around 1.8 and 0.3 respectively, and the mean around 1.2. This indicates alternative presence of small and large particles, small particles having higher frequency. The analysis of the lidar ratio values shows that in the low troposphere the extinction-to-scattering properties of aerosols varies significantly, with a preference around 48 sr. Although the majority of layers are composed of medium absorbing particles, the distribution is very wide, indicating that there are also many layers of highly absorbing aerosols, characterised by lidar ratio values above 70 sr. The median value of the lidar ratio at 355 nm in the high troposphere is very similar, 49 sr, however the spread of the values is much lower, indicating that medium absorbing particles are usually transported by the lofted layers.

Next, we analysed the aerosol types as retrieved by NATALI in high and low resolution. The overall correspondence between aerosol types in high resolution and the predominant types in low resolution is shown in Figure 7 (left panel). As explained in the methodology section, the predominant aerosol types (from left to right in the graph) may be accompanied by traces of other types (from up down in the graph), in total amount of maximum 50%. For example, NATALI identifies as predominant aerosol type "dust" if the neural networks agree that in more than 50% of the cluster points (random combinations of aerosol intensive



parameters considering the whole uncertainty interval) the values points towards "dust", while for the rest of the cluster points the neural networks may indicate, "continental" or "mineral mixtures" (in low resolution), or in addition "mixed dust", "coastal", "continental dust", "dust polluted" or "volcanic" (in high resolution). For the 63 aerosol layers for which the calibrated linear particle depolarisation was available in the RADO-Bucharest lidar datafiles, we have analysed the correspondence between aerosol types in high resolution and the predominant types in low resolution, both being provided in the NATALI output file.

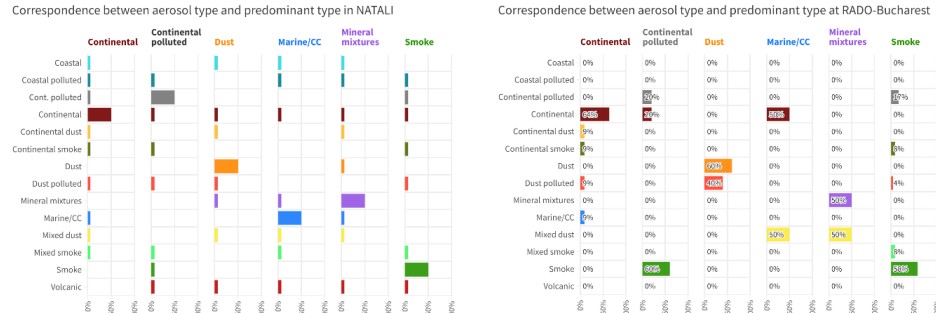

**Figure 7: Correspondence between the aerosol types retrieved in high resolution mode and the predominant aerosol type retrieved in low resolution mode: a) general correspondence in NATALI (left panel); b) correspondence as retrieved at RADO-Bucharest (right panel)**

The results show that at our location mixtures of various aerosol types are generally present. Dust is frequently mixed with smoke or industrial (identified as "dust polluted" in high resolution), continental is predominant but accompanied by small percentages of dusty mixtures ("continental dust", "dust polluted") or smoke mixtures ("continental smoke"). Mineral mixtures contain, as expected, dust and marine ("mixed dust"), smoke is accompanied by mixtures of smoke with continental and dust ("continental smoke", "mixed smoke", "dust polluted"), and marine is identified when is mixed with dust (mixed dust"). In some of the cases in low resolution, NATALI seems to attribute "marine or cloud corrupted" to continental type as identified in high resolution. Also, distinguishing continental polluted from smoke seems to be difficult in low resolution because the spectral characteristics are very similar.

Based on the cross-check between the low and high resolution typing of the 63 layers we conclude that the statistical analysis of the 229 low resolution typing can be done with a high degree of confidence, even though a small part of the "marine or cloud corrupted" cases may be in fact "continental", and a part of the "smoke" cases may be in fact "continental polluted". Figure 8 presents the altitude at which the various predominant aerosol types arrive at RADO-Bucharest (left panel) and the frequency of appearance (calculated as the ratio between the number of cases of a certain type and the total number of cases).

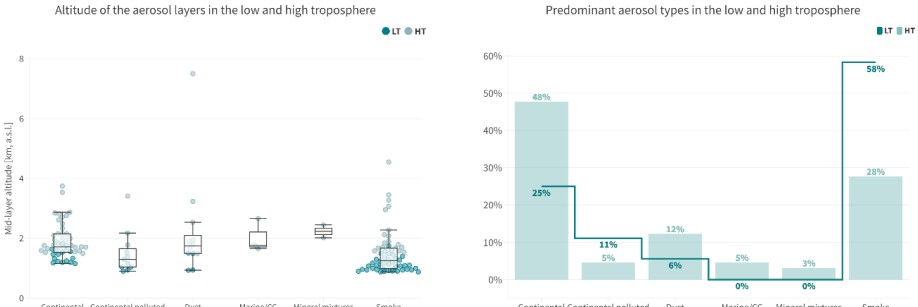

**Figure 8: Altitude (left panel) and frequency of appearance (right panel) of various aerosol predominant aerosol types**

One can note that continental and smoke aerosols are predominant at the location, with smoke more present in the low troposphere. Continental polluted and smoke particles are often observed at low altitudes (1-2 km), with several cases of long




range transported biomass burning arriving at higher altitudes (3-4 km). Dust particles are detected less often, and are two
kinds: soil dust observed in the PBL (around 1 km altitude), and mineral dust transported from desert regions arriving at higher
altitudes (2-8 km). Marine and mineral mixtures are sometimes detected around 2 km altitude. Part of these layers may be due
to long-range transported mineral dust crossing over the Black Sea or Mediterranean Sea, part may be due to incomplete cloud
screening.

Figure 9 presents the distribution of the predominant aerosol types over seasons, in the low (bottom panel) and high troposphere
(upper panel).

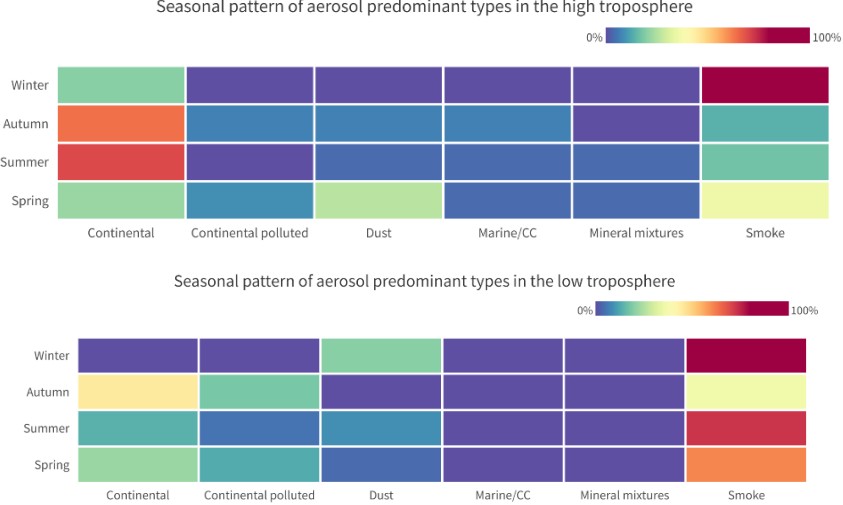

**Figure 9: Frequency of predominant aerosol types in the low troposphere (left panel) and the high troposphere (right panel)**

In the low troposphere smoke is present all seasons with a higher frequency during winter (residential heating), spring and
summer (burning of agricultural fields before sowing and after harvesting). Continental polluted particles are also present all
seasons except winter, which may be related to the wrong attribution of the type in low resolution, some of the smoke cases
during winter time being probably in fact continental pollution. Dust is detected in the low troposphere especially during
summer and winter, when more intense winds are lifting soil particles from the nearby roads and agricultural fields. During
spring and autumn wet deposition plays a role in removing the soil dust from the atmosphere. In the high troposphere smoke
is detected especially during winter, spring and summer, in relation to vegetation fires in Russia, Ukraine and Greece and
favourable circulations. Dust is observed mostly during spring when air mass transport from North Africa is frequent. During
autumn and summer, the high troposphere above our station is cleaner, continental aerosols being predominant.

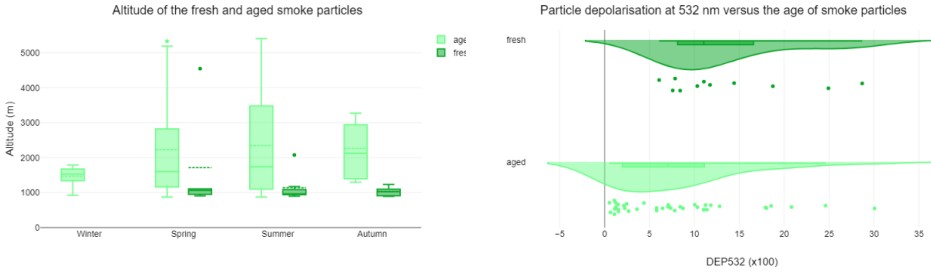



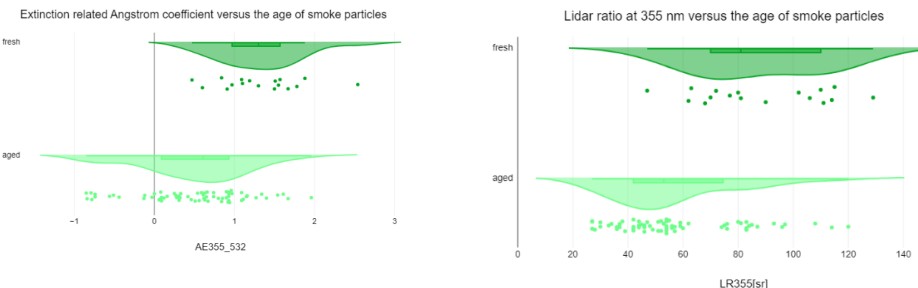

**Figure 10: Distribution, central tendency, and spread of the fresh and aged smoke characteristics: a) altitude of the smoke layers (upper left panel); b) linear particle depolarisation at 532 nm (upper right panel); c) extinction related Ängström coefficient (bottom left panel); d) lidar ratio at 355 nm (bottom right panel)**

### 3.3 FLEXPART simulations

To distinguish between the influence of local and long-range transport in the FLEXPART retro-plumes analysis, the retro-plumes trajectories (corresponding to altitude levels from 500 m to 8.0 km) were split into two clusters: below 2.0 km (corresponding to LT region) and 2.0 – 8.0 km (corresponding to HT region).

The potential sources of aerosols obtained from FLEXPART for the four seasons were classified as follows:

1. Sources that contribute to the aerosol budget at a given location and are distributed in a single region. In this category, following sources were defined: Europe, North Africa, Sahara, Middle East (Arabic Peninsula and Iran), North America (Canada and USA), Siberia.

2. Sources that contribute almost equal to the aerosol budget at a given location and are distributed in two or more regions (e.g. Sahara and Europe). These were called "Mixed" sources.

The distribution of sources by season and by the two clusters is shown in the Fig. 11.

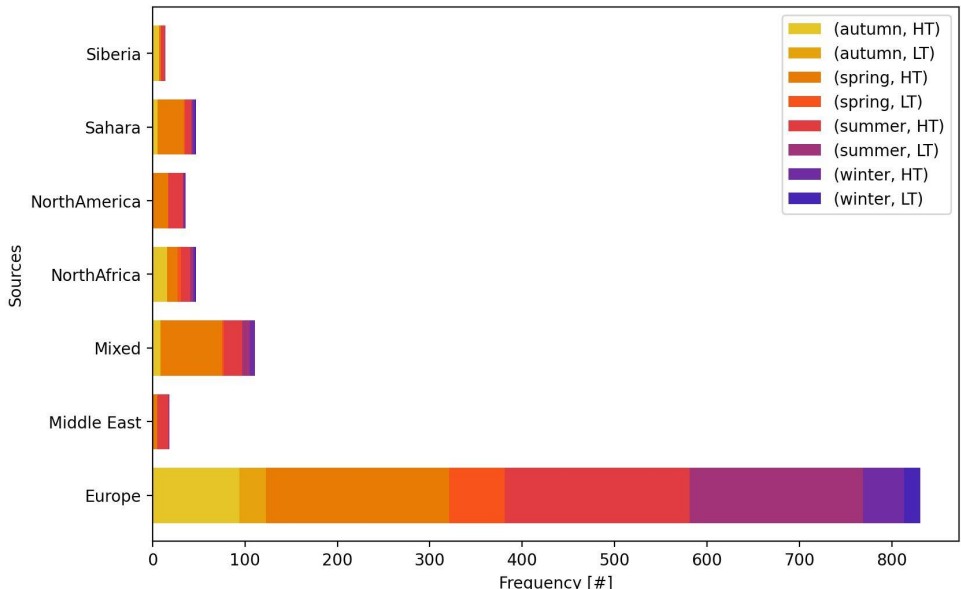

**Figure 11: The distribution of sources by season and by the LT and HT regions obtained from FLEXPART retro-plumes analysis**

As can be seen from Fig. 11, Europe is the main source of aerosols present in the entire atmospheric column for all seasons. This is followed by « Mixed » sources that contribute to the aerosol budget observed especially in the spring and summer seasons. It can also be observed that local sources as well as other sources distributed in Europe have the largest contribution



to the aerosol budget observed in LT. The contribution of sources located very far from the observation point, such as the Sahara, North Africa and North America, is poorly represented in the LT and becomes significant in the aerosol layers observed in the upper troposphere. However, in an anticyclonic type of circulation when air masses descend from the upper troposphere and reach the land surface, aerosols transported over medium- to long-distances can be observed in the lower troposphere and/or can mixed with the aerosols from PBL.

From the analysis of the FLEXPART retro-plumes, it was observed that:

• spring and summer seasons are dominated by particles emitted from local sources and aerosol particles long-range transported, such as Saharan dust or smoke from sources located in Europe or North America.

• autumn and winter seasons are dominated by particles emitted from local sources and from other sources distributed in Europe. Aerosol particles transported over long distances, such as dust from the Sahara or smoke from sources located in North Africa, have been rarely identified.

## 4    Summary and Conclusions

This paper is presenting a comprehensive analysis of aerosol properties over a ten-year period (2015–2024) at the RADO-Bucharest facility in South-East Romania. Utilizing collocated multiwavelength Raman lidar and sun/sky/lunar photometer measurements, the research focuses on retrieving and analyzing aerosol optical and microphysical properties, as well as identifying predominant aerosol types by using NATALI algorithm on multiwavelength Raman lidar data collected with the RALI system, to capture the typical properties of aerosol layers in the lower and higher atmosphere. The aerosol classification analysis of sunphotometer data revealed distinct clusters that characterize the aerosol types in the atmospheric column.

During the analyzed period, consistently low Aerosol Optical Depth (AOD) values were recorded in the atmospheric column, indicative of a relatively low aerosol burden in the atmosphere at this location. Elevated Ångström Exponent (AE) and Fine Mode Fraction (FMF) values suggest a dominance of fine-mode aerosols, typically associated with anthropogenic pollution sources. A seasonal analysis revealed a slight increase in AOD during summer months, while AE and FMF remained consistently high throughout the year, signifying persistent fine-mode aerosol prevalence. Similar results were observed over Cluj- Napoca in North-West part of Romania (Ștefănie et al., 2023) and Iasi (Cazacu et al., 2015) but also on the Romanian Black Sea Coast (Stefan et al., 2020).

Cluster analysis identified the dominant classes as Continental, Polluted, and Mixed aerosols. Episodic dust events were recorded, particularly in 2016, 2018, and 2021, corresponding to periods of intensified dust transport or enhanced local dust activity. A progressive increase in the prevalence of the Continental aerosol type was observed over the years, suggesting a gradual shift toward a more continental aerosol regime.

Analysing the aerosol layering from Lidar observations, dust events were more frequent during spring, with dust layers detected at altitudes ranging between 2 and 8 km, consistent with seasonal dust transport mechanisms, behavior also observed for Cluj area (Ștefănie et al., 2023). Biomass burning signatures were detected periodically, predominantly in July and August, coinciding with increased occurrence of wildfires over the last decade and agricultural practices of burning croplands to enhance soil conditions. Continental aerosols were more abundant during winter months, indicating a seasonal transition toward more stable, fine-mode aerosols in colder conditions.

Lidar-based aerosol layer analysis further confirmed the predominance of continental and smoke aerosols at the study site. Smoke and polluted continental aerosols were largely confined to the lower troposphere, whereas mineral dust was less frequently observed here, with soil dust present in the planetary boundary layer and long-range transported mineral dust from desert regions detected at higher altitudes. Occasional marine-mineral aerosol mixtures were identified around 2 km altitude, likely resulting from transported mineral dust crossing over maritime regions such as the Black Sea or Mediterranean Sea, as also seen sometimes over Cluj (Ștefănie et al., 2023). The NATALI algorithm analyzed 408 aerosol layers, of which 63 were





resolved at high resolution using depolarization data. High-resolution aerosol typing frequently identified layers of mixed aerosol types, such as "dust-polluted" and "continental-smoke." Lidar Ratio values exhibited variability, ranging from a median of 48 sr in the lower troposphere to 49 sr in the upper troposphere, reflecting vertical variations in aerosol optical properties,

including absorption and scattering characteristics.

Backward trajectory analysis using the FLEXPART model revealed distinct seasonal variations in aerosol source contributions consistent with the remote sensing observations. During spring and summer, aerosol concentrations were predominantly influenced by local emissions and long-range transported aerosols, including Saharan dust and biomass burning emissions from Europe and North America. Conversely, autumn and winter were characterized by a predominance of locally sourced

and European aerosols, with minimal influence from long-range transport, as Saharan dust and North African biomass smoke were rarely detected.

These findings underscore the importance of continuous, long-term aerosol monitoring with co-located lidar and photometer that allows a detailed characterization of aerosol layers, to understand temporal and spatial variations, crucial for assessing their impact on climate and air quality.


**Author contributions**

**DN**: Conceptualization, Methodology, Formal analysis, Software, Writing – original draft, Writing –

review & editing. **GAC**: Conceptualization, Formal analysis, Software, Writing –original draft. **A.N.**: Formal analysis, Investigation; Writing – review & editing. **VN**: Formal analysis, Software. **CT**: Data curation, Investigation. **JV**: Project

administration, Resources, Supervision. **AD**: Data curation, Investigation Visualization. **CR**: Data curation. **MMC**: Writing – review & editing, mentorship. **VV**: Writing – review & editing, Project administration, Resources, Supervision. **LB**: Conceptualization, Formal Analysis, Methodology, Resources, Supervision, Writing – original draft, Writing– review & editing.

**Competing interests.** One author is member of the editorial board of the Special issue Sun-photometric measurements of aerosols: harmonization, comparisons, synergies, effects, and applications (AMT/ACP inter-journal SI)

**Code/Data availability**

The NATALI (Neural Network Aerosol Typing Algorithm Based on Lidar Data) software is available with a user guide

from http://natali.inoe.ro/resources.html/software.

The AOD data from the Magurele_Inoe AERONET station, as well as the retrievals from the AERONET standard aerosol algorithm for this site, are available at https://aeronet.gsfc.nasa.gov/.

NATALI-EARLINET typing dataset is publicly available here: https://doi.org/10.57837/CNR-IMAA/ARES/NATALI-EARLINET-TYPING-2015_2023


**Acknowledgements**

This work was carried out through the Core Program within the National Research Development and Innovation Plan 2022-2027, with the support of MCID, project no. PN23 05/ 3.01.2023 and was financed by Smart Growth, Digitization and Financial Instruments Program (PoCIDIF) 2021-2027, Action 1.3 Integration of the national RDI ecosystem in the European and

international Research Space, project "Supporting the operation of facilities in Romania within the ACTRIS ERIC research infrastructure", SMIS code 309113.

Part of the work performed for this study was funded by RI-URBANS project (Research Infrastructures Services Reinforcing Air Quality Monitoring Capacities in European Urban & Industrial Areas, European Union's Horizon 2020 research and innovation program under grant agreement, contract 101036245.



The research was partially funded by the European Regional Development Fund through the Competitiveness Operational
       Programme 2014-2020, POC-A.1-A.1.1.1- F-2015, project Research Centre for Environment and Earth Observation CEO-
       Terra, SMIS code 108109, contract No.152/2016.

       Authors acknowledge AERONET-Europe for providing calibration service. AERONET-Europe is part of ACTRIS Research
       Infrastructure.

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
