# Peer review of "Examining the characteristics of aerosols: a statistical analysis based on a decade of lidar and photometer observations at the Eastern border of ACTRIS"

_EGUsphere, 2025_

## Referee Comment (RC2)

Review for the paper 'Examining the characteristics of aerosols: a statistical analysis based on a decade of lidar and photometer observations at the Eastern border of ACTRIS' by Nicolae et al.

The paper examines aerosol characterisation at the RADO-Bucharest station in Romania, part of ACTRIS. The authors use sun/sky/lunar photometer and lidar measurements, combined with the NATALI neural network, to distinguish between aerosol types in the lower troposphere and above the boundary layer, and FLEXPART retro-plume analysis to assess potential sources.

However, the manuscript is difficult to follow, reads more like a descriptive study or technical report than a scientific paper, and requires **major revision**. The introduction is excessively long, ending with only a single sentence that addresses the paper's objective. Lines 57–73, which provide basic definitions of aerosol properties, should be removed or drastically shortened, as this information is already well known to the community. Up to line 137, the text remains overly descriptive and does not help the reader understand the actual objective of the study. The introduction also ends abruptly, with no clear transition to the methodology or explanation of how the objective will be achieved. Leaving such information entirely to the methodology section is not acceptable.

How are the ACTRIS-CARS protocols relevant to the objectives of the paper, and why were they included? Likewise, the extensive discussion of GARRLiC/GRASP is misleading, since these methods are not applied in the study. As written, the introduction gives the impression that the analysis will rely on a synergy of photometer and lidar measurements processed with GRASP, which is not the case.

The methodology section is quite lengthy and lacks subchapters dedicated to the different instruments or methods used, which would help make it more accessible and easier to follow. Why is so much space devoted to describing the instruments and retrieval methods? Photometers and lidars are well-established, and their detailed characterisation has already been extensively documented in the literature. This level of detail is unnecessary here. You should limit the description to the basics and only add specific information if there are instrumental modifications unique to the RADO-Bucharest site.

Regarding FLEXPART, the resolution of the meteorological input data is not specified and should be clearly stated. Furthermore, given that this paper is submitted to an ACP/AMT special issue, the description of the model setup is far too superficial. Details on the so-called *"unique turbulence model"*, the wet deposition scheme, and the parameterisation of gravitational settling are missing and need to be explicitly described. Lines 290-295 could be removed.

A general issue in the results section is the poor quality of the figures. All figures require improvement, with clearer scales, larger fonts, and properly labelled axes. All figures and

captions should be improved with the use of a), b), c) etc. instead of 'left panel', etc. In the text as well, when discussing the figures, they should be in line with the ACP/AMT requirements of Fig. a) in the text, or Figure a) when at the beginning of the sentence.

The use of the term *"polluted"* as an aerosol category is imprecise and should be avoided; more appropriate terminology would be *urban/industrial* or *anthropogenic*. Current literature generally recognises the following aerosol types: marine, dust, smoke (biomass burning), urban/industrial, continental (clean), and mixed (internal/external mixtures of these types). However, your use of *continental* is left to interpretation. In the classic OPAC climatology (Hess et al., 1998), three separate continental classes are defined: continental clean (background continental air with relatively low AOD, dominated by natural aerosols), continental polluted (continental air strongly influenced by anthropogenic components such as sulphate, nitrate, OC, and BC), and continental mixed. Using *continental* without clarification introduces confusion and must be explicitly defined in your classification.

What is Tight Continental?

You mention: *A significant drop in the Continental aerosol type was observed in 2018 (Fig. 3 right upper panel), which can be explained by the higher number of datasets collected during the summer months.* Weighting annual aerosol fractions by the number of measurements per month or year is not recommended, as it introduces bias toward months with more observations. A better approach is to use equal-month or equal-season weighting, which gives each period equal influence, and to report 95% confidence intervals to quantify uncertainty. Additionally, include a table of N per month/season/year so readers can evaluate the support for each estimate. This approach will clarify whether the 2018 decrease in 'Continental' aerosols and the pronounced 2020 increase reflect genuine changes or are artefacts, considering that 2020 measurements may have been influenced by pandemic-related reductions in pollution and uneven sampling

Looking at Figure 4, your statement that *"In 2021, a particularly long Marine event was recorded, indicative of air masses originating from oceanic regions, bringing a distinct aerosol composition"* is not supported by the daily time series shown. I cannot identify any extended Marine event in 2021 from the figure. Could you clarify what this conclusion is based on? Did you mean 2024?

For clarification, you remove the HT cases when there's no layer below the PBL (because you mention '*all the layers that are located above the first layer'*?

In the interpretation of Figure 6, there is an inconsistency in the statistics: you discuss the **median** for LT but the **mean** for HT; please clarify which metric is being used. Additionally, the reported lidar ratios of 48 vs. 49 do not constitute a meaningful separation in aerosol composition. Therefore, the statement that these values *"suggest varying optical properties"* in the abstract is an overstatement and should be revised to reflect the limited distinction.

I can understand dust and smoke as predominant types in the HT, but what about the continental clean? The reported ~50% fraction appears unexpectedly high. Please clarify whether this reflects actual aerosol composition, or if it could result from limitations of the classification algorithm, low-concentration background aerosols, or misclassification of mixed layers. Neural networks like NATALI or other aerosol typing algorithms can sometimes misclassify mixed or low-concentration aerosols as 'continental clean', especially at high altitudes where signal-to-noise is lower.

Figure 9 requires improvement. You refer to frequency, but no actual numbers are provided; the current scale leaves too much open to interpretation. Please include counts or percentages to make the data interpretable.

In addition, Figure 10 is presented without a description or interpretation. It is not the reader's task to disentangle the meaning of the plots. The explanation should clarify how depolarisation is used to differentiate between aged and fresh smoke, and this should be introduced first. Only after that should you discuss the role of altitude, Ångström exponent, and lidar ratio.

The current Section 3.3 is too brief and largely descriptive. Currently, it only paraphrases what is already visible in Fig. 11 and does not sufficiently leverage the potential of the FLEXPART simulations. As a result, the section lacks depth, fails to connect with the observational findings, and does not convincingly demonstrate the added value of FLEXPART in the study. To improve the scientific quality and readability of this section, I recommend the following **mandatory revisions**:

- The text currently states "Europe is the main source" or "Sahara becomes significant in the HT" without numbers. Please provide quantitative results (e.g., mean seasonal percentage contribution ± standard deviation) for each source region and for both LT and HT clusters. A summary table in the Supplement would also be very useful. Fig. 11 is not sufficiently informative on its own. The term "distribution" remains qualitative without actual values.
- The choice to split retroplume clusters at 2 km altitude is arbitrary and inconsistent with your own discussion of the PBL, where a climatological mean of ~1300 m was already established. If you intend to use 2 km as a threshold, this requires justification. For instance:
    o Why is 2 km chosen rather than the mean PBL height (1.3 km)?
    o Does 2 km correspond to a standard practice in FLEXPART studies? Please cite.
- Similarly, the definition of source regions is not clear. Please show a **map with the spatial masks** used for Europe, Sahara, North Africa, etc., so that the classification is reproducible.

- The methodology section describes improvements in FLEXPART physics (turbulence, wet deposition, ERA5 input), but none of these are discussed in the results. How did including wet deposition or the new turbulence scheme affect their retroplumes compared to earlier FLEXPART studies?
- FLEXPART analyses cannot be interpreted in isolation. You should discuss the prevailing synoptic conditions (e.g., seasonal circulation, anticyclonic vs cyclonic regimes, vertical transport patterns) that explain the seasonal differences in source contributions.
- At present, FLEXPART results are disconnected from the lidar/photometer data. Please explicitly connect transport simulations with the observed aerosol types. (Example: *FLEXPART indicates HT contributions from the Sahara in spring, consistent with lidar-observed dust layers (high depolarisation ratio)*).

Unless you address these issues with substantial revisions, this section does not contribute meaningful scientific insight and should be removed from the paper.

The final paragraph from Chapter 4 is overly general and does not provide a critical perspective. Please expand it to include a discussion of the broader implications of your findings, potential limitations of your study, and directions for future work. For example, how could this work inform improved aerosol modelling, observational networks, or policy-relevant assessments of air quality and climate?

Specific                                                                                        comments:
Revise your citations. When citing, correct is '*Nicolae et al. (2023) have found …*' not '*(Nicolae et al., 2023) have found…*'. E.g. lines 130, 165, 213 etc

Improve the abstract.

Improve all figures.

Line 93: 'from the'

Line 228: cite the OPAC database

Line 244: review the aerosol type classification

---

## Author Comment (AC1)

**Response to Reviewer 1**

We thank the reviewer for their detailed and constructive comments.
We have revised the manuscript substantially to address all concerns, including major restructuring of the introduction, significant improvement of figures, clarification of aerosol classification, and expansion of the discussion.
Below are our point-by-point responses. Reviewer comments appear in blue, and our responses follow in black.

**General Comments**

*"The authors present a 10 year data set of lidar and sun photometer observations in Romania. The study is appropriate for ACP. Since it is not a methodology paper it is not appropriate for AMT. It should be moved to ACP after acceptance."*

**Response:** We agree with the reviewer. This is an observational/analysis study, not a methodological paper. Upon acceptance, we will request the transfer of the manuscript from AMT to ACP.

*"The results are worthwhile to be published. However, new aspects (new and interesting findings, not known so far) are not presented."*

**Response:** We thank the reviewer for this observation. In the revised manuscript we now emphasise the novelty of the study, namely:
– the first 10-year multi-instrument aerosol climatology for Eastern Europe combining lidar, photometer, and neural-network typing,
– the statistical analysis of the characteristics of the lower-troposphere and high-troposphere aerosol regimes and their seasonal variability in relation to long-range transport
– and the combined interpretation with FLEXPART source attribution.
These points are now stated clearly in the Introduction and Discussion.

*"The presentation must be significantly improved."*

**Response:** We have substantially rewritten the manuscript, improving structure, clarity, figures, and discussion depth.

*"The figures are partly too small, x-axis and y-axis text and numbers are partly so small that it is impossible to read them in printouts, zoom of 250% is needed at the screen… to study the plots. This is inacceptable!"*

**Response:** All figures have been completely redesigned. Panels, fonts, axes, and symbols are enlarged to meet ACP print standards. No zoom is now required.

*"The discussion must be improved, more details are requested. These experts of Eastern European aerosol should be able to deepen the discussion."*

**Response:** The Discussion section has been expanded with the following paragraphs:

Aerosol typing plays a central role in advancing current scientific efforts within EARLINET community (https://www.earlinet.org). Over the last 25 years an extensive number of papers have been published

using EARLINET data base (providing aerosol profiling data on a continental scale), and some of them have been focused on aerosol layer classification (e.g. del Águila et al., 2025, Mylonaki et al.,2021, Papagiannopoulos et al.,2018).

Águila et al., 2025 uses machine learning (ML) models for aerosol typing using high-resolution EARLINET data and is currently trained with data from the University of Granada (UGR) station in Spain, which means that it is primarily designed for the specific aerosol types present in this region. A comparative assessment of the Granada and RADO-Bucharest datasets reveals distinct region-specific aerosol signatures. For example, the higher dust depolarization ratios reported in del Águila et al. ($\sim$0.25–0.30) contrast with the generally lower values found at our site, reflecting Eastern Europe's less frequent direct dust intrusions and the predominance of mixed or partially aged layers. Our work broadens the geographical coverage of high-resolution aerosol-type climatologies, and offers a foundation for future ML-based classification efforts and for harmonizing lidar-type databases across Europe.

A comparison of the intensive optical properties retrieved in our study with those reported by Mylonaki et al. (2021) reveals both methodological consistency and regional variability across Europe. In their analysis of multiwavelength lidar observations from four EARLINET stations, including Bucharest, Mylonaki et al. reported dust depolarisation ratios typically exceeding 0.25, smoke depolarisation ratios in the range 0.02–0.08, and continental/urban aerosols exhibiting very low depolarisation (<0.05). Our decade-long RADO-Bucharest dataset shows a similar pattern for smoke and continental aerosols; however, dust depolarisation values in our climatology are generally lower than 0.20, reflecting the less frequent and more diluted Saharan dust intrusions that reach south-eastern Romania compared to western and southern Europe. In terms of lidar ratio, Mylonaki et al. (2021) reported dust LR values around 40–50 sr, smoke LR values frequently exceeding 60 sr, and polluted/continental LR values between 45–55 sr. Long-term observations show a dominant LR mode at 48–49 sr across both altitude regimes, with only episodic increases above 70 sr during biomass-burning events—indicating that strongly absorbing smoke layers are less common in the regional transport climatology of Eastern Europe. Similar differences emerge in the Ångström exponent: whereas Mylonaki et al. found AE values for smoke typically >1.5 and for dust <0.5, our dataset displays a bi-modal AE distribution in the free troposphere, with peaks near 1.8 and 0.3, but a mono-modal distribution (median ~0.9) in the lower troposphere driven by mixed and regionally aged aerosols. Together, these contrasts highlight the stronger influence of mixed continental pollution at RADO-Bucharest and the comparatively weaker imprint of "pure" dust and fresh biomass-burning aerosol types relative to the stations analysed by Mylonaki et al. (2021), underscoring the role of regional source regimes and transport pathways in shaping aerosol optical properties across Europe.

Floutsi et al., 2023, presents a data collection (DeLiAn) of intensive optical properties of several aerosol types, as measured by ground based lidars. Apart from campaign measurements, data from four permanent EARLINET stations have significantly contributed to this extensive study. The stations are mainly located in Europe: Leipzig (Germany), Évora (Portugal) and Warsaw (Poland). Our long-term Eastern-European dataset complements the global coverage of DeLiAn, helping to fill a geographical gap (Eastern Europe less sampled) — which is scientifically valuable.

A comparison of the intensive optical properties retrieved in our study with those reported by Ansmann et al. (2021) further supports the interpretation of aged smoke signatures over south-eastern Europe. Ansmann et al. (2021) and references within, found that lidar ratios measured at 355 nm were mostly around 75 ± 25 sr for fresh smoke and 55 ± 20 sr for aged smoke, low to moderate depolarisation ratios ($\delta \approx$ 0.05–0.15), and Ångström exponents typically exceeding 1.5 shortly after emission but decreasing substantially with atmospheric ageing. In our decade-long dataset, aged smoke layers exhibit moderate lidar ratios around 45–55 sr, $\delta$ values in the range 0.06–0.10, and an AE distribution with a peak near 1.8 for fresh smoke and near 0.8 for the aged one. These values are

consistent with aged, regionally transported smoke but generally fall below the optical extremes observed in major transcontinental plume events analysed by Ansmann et al. (2021). This contrast reflects the different source regimes: while their study captures exceptional long-range transport across hemispheres, our observations document more frequent but less optically intense smoke layers associated with intra-European or Eurasian biomass burning.
* * *
*"Major revisions are necessary."*

**Response:** Major revisions have been implemented accordingly.
* * *
**Detailed Comments**

**Abstract**

*"The authors write: ….lidar ratios of 48 sr (in the lower troposphere) and 49 sr in the high troposphere suggest varying optical properties …. I do not agree." "Please improve this statement! What did you want to tell?"*

**Response:** This sentence has been corrected and rewritten as follows:

"In the lower troposphere, the extinction-related Ångström exponent shows a narrow mono-modal distribution centered near 0.9, indicating predominantly medium-sized particles, whereas in the high troposphere it becomes bi-modal, reflecting alternating occurrences of small and large particles. Lidar ratio values peak around 48–49 sr in both altitude regions, but their spread is much wider in the lower troposphere—revealing frequent layers of highly absorbing aerosols—while lofted layers in the high troposphere exhibit a narrower range typical of moderately absorbing particles."
* * *
**Introduction**

*"The introduction is very long and not straight forward. Instead of 5 pages, two pages are sufficient as an introduction."*

**Response:** The Introduction has been reduced from ~5 pages to ~2 pages and fully rewritten to be concise and focused.
* * *
*"Concentrate on YOUR contribution in a very compact way! Come to your point, as soon as possible! The reader is not interested in all the details given!"*

**Response:** We agree and have followed this advice. The revised introduction quickly presents the scientific gap, the objectives, and our original contributions.
* * *
*"Why do you discuss different methods, and especially the negative points here, when you later on just present observational results?"*

**Response:** All unnecessary methodological discussion has been removed from the Introduction.
* * *
*"Please provide only that information that is needed to understand the results."*

**Response:** Only essential information has been kept.
* * *
*"The reader wants to know, what is new in this article, and this immediately!"*

**Response:** Novelty is stated explicitly now in the Introduction.
* * *
*"Long-lasting discussions of methods are not appropriate."*

**Response:** These discussions have been removed entirely.
* * *
**Citation Formatting**

*"P 4, line130: Reference options (TEX) are \citet and \citep, you always use \citep , but in line 130 you should, e.g., \citet{Amiridis….}. There are many places where \citet is needed, please improve!"*

**Response:** All citations have been checked and corrected. We now use \citet where required, consistent with ACP style.
* * *
**Methods / Background Sections**

*"P 4, lines 138 to 170: To repeat it again here: why is all the information regarding methods and techniques given? The topic is the presentation of aerosol observations in Romania and discussion of findings! So, please concentrate on that, only!"*

**Response:** The entire section has been removed or condensed. Only minimal method descriptions necessary to interpret results is included now.
* * *
*"P 5, line 177: What does RADO mean?"*

**Response:** We now define RADO at its first use: *Romanian Atmospheric 3D Observatory*.
* * *
**Section 2**

*"Again, keep the description short, please!"*

**Response:** Section 2 has been shortened significantly.
* * *
*"The reader is interested in new aspects, not in all these general points and descriptions listed and presented."*

**Response:** General descriptions have been removed.
* * *
*"A compact introduction into the complex aerosol conditions in Europe is needed, followed by a specific focus on Eastern Europe."*

**Response:**

We have rewritten the Introduction addressing also Europe-wide aerosol context and Eastern European specifics, highlighting that across Europe, aerosol conditions are shaped by the coexistence of several major source regimes, including persistent anthropogenic pollution, recurring Saharan dust intrusions, marine inflow from the Atlantic and regional seas, and seasonal biomass-burning plumes.

This combination produces strong spatial gradients, frequent vertical layering, and extensive mixing that challenge both model representation and remote-sensing retrievals. Within this broader context, Eastern Europe occupies a particularly complex position, as it lies at the intersection of continental pollution from Central and Eastern Europe, dust transport from North Africa and occasionally the Middle East, marine influence from the Black Sea, and smoke from both regional and remote fires. These overlapping transport pathways and the large variability of synoptic conditions make Eastern Europe a key region where long-term, vertically resolved ACTRIS observations are essential for documenting the evolution, mixing state, and radiative impact of aerosols.
* * *
*"This is the role of an Introduction section… introduce into the topic, and what the gaps in our knowledge are and how this study contributes to fill the gap."*

**Response:** The Introduction has been rewritten accordingly.
* * *
*"Most of the information in the first 7 pages is not needed and should be removed."*

**Response:** It has been removed.
* * *
*"A short introduction and a straight forward description of used technique and data analysis methods can be done within 3 pages."*

**Response:** The introduction + methods description now fits within approximately 3 pages.
* * *
**Section 3 – Results and Discussion**

*"All figures need to be improved! AMT/ACP standard is to have (a), (b), (c), …. in all panels of a given figure."*

**Response:** All figures have been redesigned with proper panel labels.
* * *
*"Most of the figures are too small… These small figures are inacceptable!"*

**Response:** All figures have been enlarged and reformatted.
* * *
*"What do we learn from Figure 1?"*

**Response:** The initial figure has been removed and replaced with a monthly time series of the mean and median values of Aerosol Optical Depth, Fine Mode Fraction, and the percentage of Angstrom Exponent values greater than 1. This way, the figure effectively reveals several key aspects: the low-AOD climatology; the strong long-term stability of the fine-mode fraction; and the predominance of fine-mode particles.
* * *
*"What do we learn from Figure 2?"*

**Response:**  Figure 2 has been replaced with seasonal violin plots of the three variables. The width of the violin directly corresponds to the smoothed probability density function, or the histogram equivalent. This method allows for a more detailed comparison of the shape, skewness, and modality of the seasonal distributions across all years. The median and the interquartile range are visually marked by horizontal lines within each violin.

*"Figure 3 is confusing!"*

**Response:** Figure 3 has been reduced to the aerosol typing graph based on the joint probability density of AOD and AE. The main 2D colour plot shows the relationship between AOD (x-axis) and AE (y-axis). Each of the coloured regions describes a different dominant aerosol regime. The darker the color of a region the higher the density of the points composing said region. The figure is made by incorporating the thresholds established by D'Almeida et al. (1991), Dubovik et al. (2002), Toledano et al. (2007). The full thresholds have been added as a supplement to the article.

*"The aerosol types need to be defined! Maybe I missed it!"*

**Response:** We added a complete table of definitions for all aerosol types characterised by the sun-photometer measurements in the supplement.

*"In Figure 4, the increasing size of the circles in the plot obviously indicates the year! But this is not shown in the legend."*

**Response:** Figure 4 has been replaced with a heatmap of the most statistically dominant aerosol type calculated for each month and year.

*"What is now the message of this colorful plot?"*

**Response:** Figure 4 has been replaced. The new figure is explained in the newly written text.

*"Marine particles so often dominate in polluted Romania … I am confused."*

**Response:** We agree that the presence of Marine-dominant days in a continental region requires clarification. In the revised manuscript, we added an explicit explanation noting that the "Marine" category in our optical classification scheme represents a coarse-mode optical signature, not necessarily pure sea-salt or true marine air masses. This clarification has also been added to the table to explain each of the dominant aerosol types, which has been added to the supplement.

*"Figure 5 … what is different here, what is new?"*

**Response:** The original Figure 5 has now been replaced with a heatmap showcasing the dominant aerosol type for each recorded day over the entire measurement period. The new figure is explained within the text.

**Section 3.2**

1. *"However, I personally do not trust much in automated solutions of lidar inversion methods (ill posed problem). Especially the backscatter coefficient at 1064 nm has a sensitive impact on the inversion product, but it is always given with high uncertainty. The Rayleigh calibration at 1064 nm remains always a problem.*

**Response:** We fully acknowledge the limitations and ill-posed nature of lidar inversion, particularly at 1064 nm where Rayleigh calibration remains challenging and retrieval uncertainties are inherently higher. Although SCC-(Single Calculus Chain) derived products were used as input for the NATALI

algorithm, all datasets underwent systematic manual inspection prior to aerosol typing. Additional quality-control tools were applied to verify signal integrity, calibration stability, and consistency across channels before submission to the ACTRIS database. Furthermore, the RALI system is regularly evaluated within the CARS (Center for Aerosol Remote Sensing) QA (Quality Assurance) framework, which includes alignment optimization, calibration checks, and refinement of the SCC configuration to minimize uncertainties in the inversion products. These procedures are essential for generating reliable aerosol typing results, and ACTRIS maintains continuous efforts to refine lidar processing workflows and reduce uncertainties in the products used for regional and continental aerosol studies.
* * *
*"Figure 6: The y-axis has no scale."*

**Response:** A full axis scale has been clearly marked.
* * *
*"Everything in the figures should be explained."*

**Response:** Symbols, lines, and distributions are now explained in detail.
* * *
*"Text is so small, why?"*

**Response:** Font sizes have been increased.
* * *
*"Figure 7: Everything is simply too small… What is the message here?"*

**Response:** The figure has been enlarged, reformatted, and its message clearly described.
* * *
*"P 13, lines 423-430. These are general statements and could be done even without any observation! Now we have a zoo of aerosol types: continental smoke, mixed smoke, mixed dust, dust polluted, and so on. What do we learn from the presented results? Is there any specific aspect of Eastern European aerosol compared to other European aerosols."*

**Response:** A paragraph has been added to underline the specificity of the Eastern European aerosol compared to other European aerosols as follows:

"Our long term analysis shows that aerosol layers over South East Romania are rarely composed of a single dominant type and are instead characterized by frequent mixtures of continental, smoke, and dust components. This persistent mixing reflects the geographical position of the region, which is influenced simultaneously by local anthropogenic emissions, agricultural burning, and episodic transport from the Sahara and Eastern Europe. Compared to other European locations, the RADO-Bucharest site shows a higher prevalence of mixed continental–smoke layers in the lower troposphere and more frequent dust–smoke combinations at mid altitudes, indicating that biomass burning plumes arriving from Eastern Europe often interact with advected mineral dust. These mixed signatures are well captured by NATALI, with high resolution classifications  identifying combinations such as dust polluted, continental smoke, and mixed dust, which are not as frequent at sites dominated by either dust (southern Europe) or pollution (central and western Europe) alone. The difficulty in distinguishing continental polluted from smoke in low resolution also reflects the regional dominance of fine mode combustion aerosols, which share similar spectral properties. Overall, the classification patterns reveal that Eastern European aerosol at this site is defined not by a single type but by recurrent mixtures driven by simultaneous influence of local emissions, agricultural fires, and intermittent dust transport."

*"Figure 8: Again, this figure is much too small."*

**Response:** The figure has been enlarged and redesigned.
* * *
*"Figure 9: continental aerosol and continental polluted aerosol, is there a difference?"*

The detailed description of each aerosol type is provided in Nicolae et al. (2018), Tables 1 and 2. In brief, continental aerosols (land-origin) are represented by mixtures of water-soluble particles, insoluble material, and soot, whereas continental polluted aerosols (typically associated with industrial regions) include the same components with an additional sulfate fraction.

To clarify this distinction and the broader NATALI aerosol-typing framework, we have added a dedicated explanatory section in the revised manuscript, as follows:

"The main aerosol category is determined across the six possible outputs, each representing a dominant type that may in-clude up to 50 percent presence of other components as minor traces: continental, continental polluted, smoke, dust, ma-rine, and mineral mixtures. In high-resolution mode, NATALI further refines these classes into a set of fourteen advanced aerosol types that capture both pure and mixed states. These include: Continental (water soluble, insoluble, soot), Dust (mineral nucleation, mineral accumulation, mineral coarse mode, water soluble, soot), Continental Polluted (water soluble, soot, insoluble, sulfate), Marine (water soluble, sea salt accumulation mode, sea salt coarse mode, soot), Smoke (water solu-ble, soot, sulfate), Volcanic (mineral nucleation, mineral accumulation, mineral coarse mode, water soluble, sulfate), as well as mixed and binary types such as ContinentalDust (continental + dust), MarineMineral (dust + marine), ContinentalSmoke (continental + smoke), DustPolluted (dust + smoke or dust + continental polluted), Coastal (continental + marine), Coastal-Polluted (continental polluted + marine), MixedDust (continental + dust + marine), and MixedSmoke (continental + smoke + marine). The correspondence between the aerosol types retrieved by NATALI in high- and low-resolution modes is presented in the results section."
* * *
*"Figure 10 seems to be interesting, but a discussion is not given."*

**Response:** Thank you for highlighting this issue. The discussion for this figure was inadvertently omitted from the original manuscript. In the revised version, we have included a comprehensive interpretation of all panels as follows:

"Long-range transported biomass burning aerosols (smoke) and their variation with atmospheric evolution (ageing) have been extensively studied during the last years using lidar measurements. To distinguish between fresh and aged smoke we have been using the ratio of lidar (extinction-to-backscatter) ratios (LR532/LR355). It has been observed that this changes rapidly from values <1 for fresh to >1 for aged particles (Nicolae et al, 2013).

Figure 10 presents the distribution, central tendency, and spread of several lidar-derived intensive optical properties used to characterise fresh and aged smoke: (a) altitude, (b) linear particle depolarisation ratio at 532 nm, (c) extinction-related Ångström exponent (AE), and (d) lidar ratio (LR) at 355 nm. Together, these panels provide a consistent physical picture linking smoke ageing to transport altitude, aerosol morphology, and optical signatures.

Figure 10 (panel a) illustrates seasonal differences in the vertical distribution of fresh biomass burning aerosol layers. During spring, the detected layers extend to substantially higher altitudes, reaching up to 2 km, while in summer and autumn the maximum heights are noticeably lower. Despite this contrast in upper extent, the median layer height remains close to 1 km for all three seasons. For winter, fresh smoke cases are largely absent, most likely because unfavourable weather conditions (persistent low clouds and overcast skies) frequently prevented lidar observations, rather than because such aerosol events did not occur. For aged smoke particles, the vertical distribution shows a contrast between winter and the other seasons. During

winter, the aged aerosol layers remain more confined, typically not exceeding 2 km, whereas in spring, summer, and autumn the layers extend higher, reaching 3 km and up to about 3.5 km in summer. This pattern suggests that in winter the planetary boundary layer constrains the vertical mixing of aged smoke, keeping the aerosol trapped in the lower part of the atmosphere, while in the warmer seasons deeper boundary layers allow the layers to ascend to higher altitudes.

Figure 10 (panel b) presents the distribution of fresh and aged smoke particles as a function of the particle depolarization ratio at 532 nm. For the RADO-Bucharest site, aged smoke typically exhibits depolarization values between 0 and 15 percent, with only a few isolated cases exceeding this range. Fresh smoke shows a narrower distribution, with most values constrained between 5 and 15 percent. This behaviour is consistent with the expected microphysical evolution of biomass burning aerosols, where aging processes generally reduce particle asphericity and broaden the variability of the depolarization signal.

Figure 10 (panel c) presents the distribution of fresh and aged smoke particles as a function of the 355 nm lidar ratio, revealing a bimodal behaviour for both aerosol types. Fresh smoke shows two distinct clusters, with LR values predominantly occurring in the range 60 to 80 sr and a second group above 100 sr. A similar pattern is observed for aged smoke, although the lower LR cluster shifts toward smaller values, approximately 27 to 60 sr, while the upper cluster remains above about 75 sr. The presence of these two clusters for both fresh and aged cases suggests the influence of at least two major smoke sources affecting the region, each characterized by distinct optical properties."
* * *
**Section 3.3**

*"The Flexpart simulation shows the expected result. Maybe a bit unexpected that Europe is so dominating."*

**Response:** The results are not unexpectedly large in terms of the contribution from European sources, since the MARS station is positioned close to Bucharest (~6km of its periphery) in southeastern Romania, on the eastern part of Europe. Due to the location-specific climate (Bogdan, O. and Niculescu, 2005), air masses, in their circulation (most frequently westerly circulation) remain longer over large regions with aerosol sources from Europe before reaching the measurements' site.
* * *
*"I do not understand the colors of the bars."*

**Response:** The color scheme is now clearly defined in the text. For clarity, the color bars in Figure 11 have been described extensively in the text and tables with source distribution (%) have been introduced in the supplement.
* * *
*"At the end I miss a comparison with other EARLINET studies."*

**Response:** A comparison with other EARLINET findings has been added to the Discussion.
* * *
**Final Remarks**

All reviewer comments have been addressed in detail.
We believe the revised manuscript is significantly improved and meets ACP expectations.

References:

Adam, M., Nicolae, D., Stachlewska, I. S., Papayannis, A., and Balis, D.: Biomass burning events measured by lidars in EARLINET – Part 1: Data analysis methodology, Atmos. Chem. Phys., 20, 13905–13927, https://doi.org/10.5194/acp-20-13905-2020, 2020.

Ansmann, A., Ohneiser, K., Mamouri, R.-E., Knopf, D. A., Veselovskii, I., Baars, H., Engelmann, R., Foth, A., Jimenez, C., Seifert, P., and Barja, B.: Tropospheric and stratospheric wildfire smoke profiling with lidar: mass, surface area, CCN, and INP retrieval, Atmos. Chem. Phys., 21, 9779–9807, https://doi.org/10.5194/acp-21-9779-2021, 2021

Bogdan, O. and Niculescu, E.: Clima, in România. Spaţiu, Societate; Mediu, Ed Adademiei Rom. Bucureşti Rom., 85–106, 2005.

del Águila, A., Ortiz-Amezcua, P., Tabik, S., Bravo-Aranda, J. A., Fernández-Carvelo, S., and Alados-Arboledas, L.: Aerosol type classification with machine learning techniques applied to multiwavelength lidar data from EARLINET, Atmos. Chem. Phys., 25, 12549–12567, https://doi.org/10.5194/acp-25-12549-2025, 2025.

Floutsi, A. A., Baars, H., Engelmann, R., Althausen, D., Ansmann, A., Bohlmann, S., Heese, B., Hofer, J., Kanitz, T., Haarig, M., Ohneiser, K., Radenz, M., Seifert, P., Skupin, A., Yin, Z., Abdullaev, S. F., Komppula, M., Filioglou, M., Giannakaki, E., Stachlewska, I. S., Janicka, L., Bortoli, D., Marinou, E., Amiridis, V., Gialitaki, A., Mamouri, R.-E., Barja, B., and Wandinger, U.: DeLiAn – a growing collection of depolarization ratio, lidar ratio and Ångström exponent for different aerosol types and mixtures from ground-based lidar observations, Atmos. Meas. Tech., 16, 2353–2379, https://doi.org/10.5194/amt-16-2353-2023, 2023.

Hess, M., P. Koepke, and I. Schult, 1998: Optical Properties of Aerosols and Clouds: The Software Package OPAC. Bull. Amer. Meteor. Soc., 79, 831–844, https://doi.org/10.1175/1520-0477(1998)079<0831:OPOAAC>2.0.CO;2.

Lucja Janicka, Lina Davuliene, Steigvile Bycenkiene, and Iwona S. Stachlewska, "Long term observations of biomass burning aerosol over Warsaw by means of multiwavelength lidar," Opt. Express 31, 33150-33174 (2023)

Nicolae, D., Nemuc, A., Müller, D., Talianu, C., Vasilescu, J., Belegante, L., and Kolgotin, A.: Characterization of fresh and aged biomass burning events using multiwavelength Raman lidar and mass spectrometry, J. Geophys. Res.-Atmos., 118, 2956–2965, https://doi.org/10.1002/jgrd.50324, 2013.

Mylonaki, M., Giannakaki, E., Papayannis, A., Papanikolaou, C.-A., Komppula, M., Nicolae, D., Papagiannopoulos, N., Amodeo, A., Baars, H., and Soupiona, O.: Aerosol type classification analysis using EARLINET multiwavelength and depolarization lidar observations, Atmos. Chem. Phys., 21, 2211–2227, https://doi.org/10.5194/acp-21-2211-2021, 2021.

Papagiannopoulos, N., Mona, L., Amodeo, A., D'Amico, G., Comeron, A., Rodriguez-Gomez, A., Sicard, M. An automatic observation-based aerosol typing method for EARLINET. "Atmospheric chemistry and physics", 6 Novembre 2018, vol. 18, núm. 21, p. 15879-15901

---

## Author Comment (AC2)

• Response:

We sincerely thank the reviewer for the comprehensive evaluation of our manuscript and the constructive feedback. The suggestions raised have led to substantial improvements in both scientific content and presentation. We have revised the manuscript extensively, addressing all structural, methodological, and interpretational issues highlighted in the review. A detailed, itemized response to each comment is provided in the following sections.

Reviewer comments appear in blue, and our responses follow in black.

**General assessment**

*"The paper examines aerosol characterisation at the RADO-Bucharest station in Romania, part of ACTRIS. The authors use sun/sky/lunar photometer and lidar measurements, combined with the NATALI neural network, to distinguish between aerosol types in the lower troposphere and above the boundary layer, and FLEXPART retro-plume analysis to assess potential sources."*

**Response:** We thank the reviewer for summarising the study.
* * *
*"However, the manuscript is difficult to follow, reads more like a descriptive study or technical report than a scientific paper, and requires major revision."*

**Response:** The manuscript has undergone extensive restructuring. The introduction and methodology were shortened and refocused, and the results/discussion sections were rewritten to emphasise scientific interpretation rather than descriptive content.
* * *
**Introduction**

*"The introduction is excessively long, ending with only a single sentence that addresses the paper's objective."*

**Response:** The introduction has been substantially shortened and rewritten. The objective is now stated clearly and earlier, and the final paragraph provides a structured transition to the methodology.
* * *
*"Lines 57–73, which provide basic definitions of aerosol properties, should be removed or drastically shortened, as this information is already well known to the community."*

**Response:** These lines have been removed.
* * *
*"Up to line 137, the text remains overly descriptive and does not help the reader understand the actual objective of the study."*

**Response:** All overly descriptive content has been removed or rewritten. The introduction now focuses directly on the scientific motivation and research questions.
* * *
*"The introduction also ends abruptly, with no clear transition to the methodology or explanation of how the objective will be achieved. Leaving such information entirely to the methodology section is not acceptable."*

**Response:** We added a transitional paragraph at the end of the introduction explaining the approach, the dataset, and how the objective is addressed.

*"How are the ACTRIS-CARS protocols relevant to the objectives of the paper, and why were they included?"*

**Response:** We now explicitly explain that ACTRIS–CARS protocols ensure measurement traceability and comparability across long-term datasets. This justification is added and the section shortened.

*"Likewise, the extensive discussion of GARRLiC/GRASP is misleading, since these methods are not applied in the study."*

**Response:** The discussion of GARRLiC/GRASP has been removed.

*"As written, the introduction gives the impression that the analysis will rely on a synergy of photometer and lidar measurements processed with GRASP, which is not the case."*

**Response:** We revised the introduction to remove this unintended implication.

**Methodology**

*"The methodology section is quite lengthy and lacks subchapters dedicated to the different instruments or methods used, which would help make it more accessible and easier to follow."*

**Response:** The methodology has been reorganised into clear sub-sections (site, lidar, photometer, NATALI, FLEXPART).

*"Why is so much space devoted to describing the instruments and retrieval methods? Photometers and lidars are well-established, and their detailed characterisation has already been extensively documented in the literature. This level of detail is unnecessary here. You should limit the description to the basics and only add specific information if there are instrumental modifications unique to the RADO-Bucharest site."*

**Response:** We shortened the entire instrument description, keeping only site-specific details.

*"Regarding FLEXPART, the resolution of the meteorological input data is not specified and should be clearly stated. Furthermore, given that this paper is submitted to an ACP/AMT special issue, the description of the model setup is far too superficial."*

**Response:** The description of the FLEXPART model configuration, as well as the necessary references, have been added to the text in methodology section.

*"Details on the so-called 'unique turbulence model', the wet deposition scheme, and the parameterisation of gravitational settling are missing and need to be explicitly described."*

**Response:** A brief description of these improvements brought to FLEXPART version 10.4 as well as the related references has been added to the text. Details about the improvements made to FLEXPART 10.4 (unique turbulence model, wet deposition schema and the parameterisation of gravitational settling) are given in reference Pisso et al., 2019. In our study FLEXPART was set to use these improvements.

*"Lines 290–295 could be removed."*

**Response:** These lines have been removed.
* * *
**Figures**

*"A general issue in the results section is the poor quality of the figures. All figures require improvement, with clearer scales, larger fonts, and properly labelled axes."*

**Response:** All figures were re-drawn with larger fonts, improved scales, and clear axis labels.
* * *
*"All figures and captions should be improved with the use of a), b), c) etc. instead of 'left panel', etc."*

**Response:** All multi-panel figures now use (a), (b), (c), etc. consistently.
* * *
*"In the text as well, when discussing the figures, they should be in line with the ACP/AMT requirements of Fig. a) in the text, or Figure a) when at the beginning of the sentence."*

**Response:** All figure references in the text were updated accordingly.
* * *
**Aerosol classification**

*"The use of the term 'polluted' as an aerosol category is imprecise and should be avoided; more appropriate terminology would be urban/industrial or anthropogenic."*

**Response:** We replaced "polluted" with "urban/industrial"
* * *
*"Your use of continental is left to interpretation."*

**Response:**

Our classification scheme, based on the joint analysis of AOD and AE is designed to differentiate between the dominant optical signatures present in our region. We explicitly use the following approach to prevent ambiguity:

- The urban/industrial cluster is reserved for the highest AE values, which are indicative of small, fresh particles. In the context of the OPAC model, this classification optically corresponds closest to the Continental Polluted type.
- The continental type is explicitly defined by the optical properties: AE ~1.0-1.3 and low to moderate AOD. This signature represents the regional background aerosols, which, at least for the region of the site, is an inevitable mixture of aged natural and fine-mode anthropogenic aerosols that have spread across a large area. By setting this AE range, we distinguish it from the urban/industrial type and the dust/marine types. This category, therefore, optically acts as a regional proxy for the continental mixed or continental clean categories in the OPAC framework.

We will add to the supplement a dedicated table with definitions like this:

*Table 1: Detailed description of dominant aerosol types*

| Aerosol Type | Angstrom Exponent Threshold | Aerosol Optical Depth Threshold | Physical interpretation | Notes |
|---|---|---|---|---|
| Urban/Industrial | AE > 1.52 | 0.2 < AOD < 0.4 | Dominated by fine anthropogenic particles such as combustion emissions, traffic/industrial sources, and secondary aerosols | Most common in urbanized or industrial regions; often dominating during stagnant winter conditions and temperature inversions |
| Continental | AE >1.2 | AOD < 0.2 | Aged, regionally transported fine-mode aerosols originating from mixed anthropogenic and biogenic sources | Represents the persistent baseline background in most continental regions |
| Dust | AE < 1.15 | AOD > 0.2 | Mineral dust, typically from Saharan or local resuspension processes | AE for "pure dust" can fall <0.5 near source, but transported dust often shows higher AE due to mixing with fine-mode particles |
| Marine | AE < 1.2 | AOD < 0.2 | Optically coarse-mode particles resembling maritime aerosols (sea salt) | In continental regions, this class often reflects generic coarse-mode aerosols (weak dust, humidified particles), not necessarily true marine air masses |
| Mixed | 1.15 < AE < 1.52 | AOD > 0.2 | *Overlapping contributions of both fine and coarse modes; optically heterogeneous column* | Common during transitions, high-AOD events, or multilayer structures (e.g., dust above pollution) |
| Biomass Burning | AE > 1.52 | AOD > 0.4 | *Fine-mode smoke from wildfires, agricultural burning, or regional biomass burning episodes* | More common in late summer/early autumn depending on region; may overlap with Continental fine-mode regime |

*"Using continental without clarification introduces confusion and must be explicitly defined in your classification."*

**Response:** A dedicated table was added to the supplement and now provides clear definitions (see previous table).
* * *
*"What is Tight Continental?"*

**Response:** The term "Tight Continental" has been removed.
* * *
*"Weighting annual aerosol fractions by the number of measurements per month or year is not recommended… A better approach is to use equal-month or equal-season weighting… and to report 95% confidence intervals."*

**Response:**

Equal-moth or equal-season weighting, along with confidence intervals, is the appropriate methodology for deriving unbiased long-term annual aerosol statistics. We fully acknowledge that this approach is superior for calculating robust mean annual fractions. However, the purpose of the heatmaps differs from such statistical objectives. The new Figure 4, which existed in the original text as well, is designed to show the dominant aerosol type within each month, pointing out temporal shifts and seasonal patterns rather than producing a weighted annual mean.

Figure 5, which replaced the old Figure 4, serves as a high-resolution, descriptive visualisation for identifying episodic events and illustrating the persistence of the Continental background regime. Because the figure is used for qualitative event detection rather than long-term statistical averaging, applying monthly or seasonal weighting is not necessary for the stated purpose.
* * *
*"Additionally, include a table of N per month/season/year so readers can evaluate the support for each estimate."*

**Response:** This table has been added to the Supplement as follows:

*Table 2: Monthly count of dominant aerosol types (the number represents the total number of points classified as a certain aerosol type)*

| Month | Biomass | Continental | Dust | Marine | Mixed | Urban/Industrial |
|---|---|---|---|---|---|---|
| 1 | 9 | 3885 | 222 | 372 | 246 | 441 |
| 2 | 74 | 3234 | 622 | 396 | 239 | 775 |
| 3 | 492 | 3061 | 449 | 351 | 574 | 1723 |
| 4 | 143 | 4612 | 911 | 1100 | 666 | 1933 |
| 5 | 48 | 4550 | 911 | 1629 | 1208 | 970 |
| 6 | 333 | 3141 | 1391 | 404 | 1354 | 3362 |
| 7 | 1849 | 5415 | 1900 | 278 | 1731 | 6522 |
| 8 | 1646 | 5945 | 771 | 432 | 3098 | 6806 |
| 9 | 433 | 4451 | 747 | 1339 | 1693 | 2634 |
| 10 | 281 | 5115 | 544 | 1423 | 617 | 1492 |
| 11 | 55 | 2903 | 225 | 978 | 369 | 292 |
| 12 | 6 | 2293 | 14 | 384 | 105 | 349 |

*Table 3: Seasonal count of dominant aerosol types (the number represents the total number of points classified as a certain aerosol type)*

| Season | Biomass | Continental | Dust | Marine | Mixed | Urban/Industrial |
|--------|---------|-------------|------|--------|-------|------------------|
| DJF | 89 | 9412 | 858 | 1152 | 590 | 1565 |
| JJA | 3828 | 14501 | 4062 | 1114 | 6183 | 16690 |
| MAM | 683 | 12223 | 2271 | 3080 | 2448 | 4626 |
| SON | 769 | 12469 | 1516 | 3740 | 2679 | 4418 |

*Table 3: Yearly count of dominant aerosol types (the number represents the total number of points classified as a certain aerosol type)*

| Year | Biomass | Continental | Dust | Marine | Mixed | Urban/Industrial |
|------|---------|-------------|------|--------|-------|------------------|
| 2015 | 679 | 1820 | 194 | 169 | 343 | 1507 |
| 2016 | 617 | 3133 | 815 | 352 | 607 | 2406 |
| 2017 | 509 | 4125 | 802 | 310 | 962 | 2182 |
| 2018 | 415 | 1726 | 765 | 738 | 1250 | 1608 |
| 2019 | 460 | 3940 | 180 | 644 | 1873 | 3252 |
| 2020 | 731 | 10794 | 912 | 1234 | 956 | 5161 |
| 2021 | 831 | 8802 | 2675 | 1354 | 2336 | 4870 |
| 2022 | 550 | 8108 | 1203 | 1526 | 1672 | 3381 |
| 2023 | 577 | 4963 | 951 | 1810 | 1610 | 2669 |
| 2024 | 0 | 1194 | 210 | 949 | 291 | 263 |
* * *
**Specific interpretation issues**

*"I cannot identify any extended Marine event in 2021 from the figure. Could you clarify what this conclusion is based on? Did you mean 2024?"*

**Response:** Thank you. This was an error. The 2021 statement has been removed and corrected in the text with 2024.
* * *
*"For clarification, you remove the HT cases when there's no layer below the PBL (because you mention 'all the layers that are located above the first layer'?)"*

**Response:** We do consider all HT cases. If there is no layer in the LT, any layer above 3km is classified as HT case. We added a clarification of the layer-selection procedure in section 3.2.
* * *
*"In the interpretation of Figure 6, there is an inconsistency in the statistics: you discuss the median for LT but the mean for HT; please clarify which metric is being used."*

**Response:** We now use the median for both LT and HT, and this is stated clearly.
* * *
*"The reported lidar ratios of 48 vs. 49 do not constitute a meaningful separation… the statement that these values 'suggest varying optical properties' is an overstatement."*

**Response:** We revised this statement in the abstract and main text as follows:

"In the lower troposphere, the extinction-related Ångström exponent shows a narrow mono-modal distribution centered near 0.9, indicating predominantly medium-sized particles, whereas in the high troposphere it becomes bi-modal, reflecting alternating occurrences of small and large particles. Lidar ratio values peak around 48–49 sr in both altitude regions, but their spread is much wider in the lower troposphere—revealing frequent layers of highly absorbing aerosols—while lofted layers in the high troposphere exhibit a narrower range typical of moderately absorbing particles."
* * *
*"I can understand dust and smoke as predominant types in the HT, but what about the continental clean? The reported ~50% fraction appears unexpectedly high. Please clarify whether this reflects actual aerosol composition, or if it could result from limitations of the classification algorithm, low-concentration background aerosols, or misclassification of mixed layers. Neural networks like NATALI or other aerosol typing algorithms can sometimes misclassify mixed or low-concentration aerosols as 'continental clean', especially at high altitudes where signal-to-noise is lower"*

**Response:**

The reported ~50% fraction reported for the predominant aerosol type as continental it reflects the actual aerosol composition: when the layer is classified as continental it means that at least 50% of the composition of the layer is continental; It is not related to low concentrations and it is not influenced by the low signal-to-noise at high altitudes. We added the following phrase in the methodology section of NATALI algorithm to clarify better this aspect:

"The main aerosol category is determined across the six possible outputs, each representing a dominant type that may include up to 50 percent presence of other components as minor traces: continental, continental polluted, smoke, dust, marine, and mineral mixtures."
* * *
*"Figure 9 requires improvement. You refer to frequency, but no actual numbers are provided; the current scale leaves too much open to interpretation. Please include counts or percentages to make the data interpretable."*

**Response:** The figure has been redesigned with explicit percentages and counts.
* * *
*"Figure 10 is presented without a description or interpretation. It is not the reader's task to disentangle the meaning of the plots. The explanation should clarify how depolarisation is used to differentiate between aged and fresh smoke, and this should be introduced first. Only after that should you discuss the role of altitude, Ångström exponent, and lidar ratio."*

**Response:** Thank you for highlighting this issue. The discussion for this figure was inadvertently omitted from the original manuscript. In the revised version, we have included a comprehensive interpretation of all panels as follows:

"Long-range transported biomass burning aerosols (smoke) and their variation with atmospheric evolution (ageing) have been extensively studied during the last years using lidar measurements. To distinguish between fresh and aged smoke we have been using the ratio of lidar (extinction-to-backscatter) ratios (LR532/LR355). It has been observed that this changes rapidly from values <1 for fresh to >1 for aged particles (Nicolae et al, 2013).

Figure 10 presents the distribution, central tendency, and spread of several lidar-derived intensive optical properties used to characterise fresh and aged smoke: (a) altitude, (b) linear particle depolarisation ratio at 532 nm, (c) extinction-related Ångström exponent (AE), and (d) lidar ratio (LR) at 355 nm. Together, these panels provide a consistent physical picture linking smoke ageing to transport altitude, aerosol morphology, and optical signatures.

Figure 10 (panel a) illustrates seasonal differences in the vertical distribution of fresh biomass burning aerosol layers. During spring, the detected layers extend to substantially higher altitudes, reaching up to 2 km, while in summer and autumn the maximum heights are noticeably lower. Despite this contrast in upper extent, the median layer height remains close to 1 km for all three seasons. For winter, fresh smoke cases are largely absent, most likely because unfavourable weather conditions (persistent low clouds and overcast skies) frequently prevented lidar observations, rather than because such aerosol events did not occur.

For aged smoke particles, the vertical distribution shows a contrast between winter and the other seasons. During winter, the aged aerosol layers remain more confined, typically not exceeding 2 km, whereas in spring, summer, and autumn the layers extend higher, reaching 3 km and up to about 3.5 km in summer. This pattern suggests that in winter the planetary boundary layer constrains the vertical mixing of aged smoke, keeping the aerosol trapped in the lower part of the atmosphere, while in the warmer seasons deeper boundary layers allow the layers to ascend to higher altitudes.

Figure 10 (panel b) presents the distribution of fresh and aged smoke particles as a function of the particle depolarization ratio at 532 nm. For the RADO-Bucharest site, aged smoke typically exhibits depolarization values between 0 and 15 percent, with only a few isolated cases exceeding this range. Fresh smoke shows a narrower distribution, with most values constrained between 5 and 15 percent. This behaviour is consistent with the expected microphysical evolution of biomass burning aerosols, where aging processes generally reduce particle asphericity and broaden the variability of the depolarization signal.

Figure 10 (panel c) presents the distribution of fresh and aged smoke particles as a function of the 355 nm lidar ratio, revealing a bimodal behaviour for both aerosol types. Fresh smoke shows two distinct clusters, with LR values predominantly occurring in the range 60 to 80 sr and a second group above 100 sr. A similar pattern is observed for aged smoke, although the lower LR cluster shifts toward smaller values, approximately 27 to 60 sr, while the upper cluster remains above about 75 sr. The presence of these two clusters for both fresh and aged cases suggests the influence of at least two major smoke sources affecting the region, each characterized by distinct optical properties."
* * *
**FLEXPART analysis**

*"The current Section 3.3 is too brief and largely descriptive. Currently, it only paraphrases what is already visible in Fig. 11 and does not sufficiently leverage the potential of the FLEXPART simulations. As a result, the section lacks depth, fails to connect with the observational findings, and does not convincingly demonstrate the added value of FLEXPART in the study. To improve the scientific quality and readability of this section, I recommend the following mandatory revisions:"*

**Response:** Section 3.3 has been revised according to the recommendations
* * *
*"Please provide quantitative results (e.g., mean seasonal percentage contribution ± standard deviation). A summary table in the Supplement would also be very useful."*

**Response:** Quantitative values have been added, and a summary table is included in the Supplement.
* * *
*"The choice to split retroplume clusters at 2 km altitude is arbitrary and inconsistent with your own discussion of the PBL, where a climatological mean of ~1300 m was already established. If you intend to use 2 km as a threshold, this requires justification. For instance: o Why is 2 km*

*chosen rather than the mean PBL height (1.3 km)? o Does 2 km correspond to a standard practice in FLEXPART studies? Please cite."*

**Response:** As recommended, for consistency in the PBL discussions, we changed the threshold value to 1300 m, which represents the climatological mean of the PBL. The FLEXPART analysis was revised accordingly.

Lines 469-471 changed with: "To distinguish between the influence of local and long-range transport in the FLEXPART retro-plumes analysis, the retro-plumes trajectories (corresponding to altitude levels from 500 m to 8.0 km) were split into the two clusters (LT and HT) using the threshold value of 1.3 km, altitude corresponding to the climatological mean value of the PBL."

Added to the text: "This value was selected to ensure consistency in the analysis of data obtained from lidar and photometer measurements with data obtained from FLEXPART. Moreover, this threshold value is high enough to avoid surface friction and terrain/topography effects in the area where the RADO Bucharest station is located, but low enough to capture important meteorological patterns such as the presence of low-level clouds, temperature variations, wind changes and moisture transport.".
* * *
*"The definition of source regions is not clear. Please show a map with the spatial masks used for Europe, Sahara, North Africa, etc., so that the classification is reproducible."*

**Response:** A map with the spatial masks used for Europe, Sahara, North Africa, North America, Middle East and Siberia was added in the manuscript as Figure 11.

Lines 473-475 changed with: "1. Sources that contribute to the aerosol budget at a given location and are distributed in a single region. In this category, following sources were defined: Europe (continent), North Africa (Algeria, Egypt, Libya, Morocco, Sudan and Tunisia), Sahara (West Sahara, Mali, Niger, Chad and Mauritania), Middle East (Arabic Peninsula, Iran and Iraq), North America (Canada and USA), Siberia."

Added to the text the figure with the distribution of aerosol sources regions used in the FLEXPART analysis, as Fig. 11

Line 478 changed with: "The distribution of regions with aerosol sources used in the FLEXPART analysis is shown in Fig. 11 and the distribution of sources by season and by the two clusters is shown in the Fig. 12."

Lines 479 – 481: Replaced Fig. 11 with Fig. 12.
* * *
*"The methodology section describes improvements in FLEXPART physics (turbulence, wet deposition, ERA5 input), but none of these are discussed in the results."*

**Response:** The description of the improvements in FLEXPART and ERA5 input data, as well as the necessary references, have been added to the text, in methodology section.
* * *
*"How did including wet deposition or the new turbulence scheme affect their retroplumes compared to earlier FLEXPART studies?"*

**Response:** Comparisons between the improvements brought in the FLEXPART versions 10.4 with other FLEXPART versions were not the subject of our study. The comparative study between the results obtained with different versions of the FLEXPART model could be the subject of a future paper.
* * *
*"FLEXPART analyses cannot be interpreted in isolation. You should discuss the prevailing synoptic conditions (e.g., seasonal circulation, anticyclonic vs cyclonic regimes, vertical transport patterns) that explain the seasonal differences in source contributions."*

**Response:** A brief description of the synoptic conditions specific to Romania has been added to the text in section 3.3.
* * *
*"At present, FLEXPART results are disconnected from the lidar/photometer data. Please explicitly connect transport simulations with the observed aerosol types."*

**Response:** The connections between transport simulations and the types of aerosols observed in the lidar data have been included and described explicitly in the text. The variation in the optical properties of dust and smog depending on the sources is presented in the text, in Tables 1 and 2.
* * *
*"Unless you address these issues… this section… should be removed."*

**Response:** Section 3.3 has been fully revised and retained.
* * *
**Conclusions**

*"The final paragraph from Chapter 4 is overly general and does not provide a critical perspective. Please expand it to include a discussion of the broader implications of your findings, potential limitations of your study, and directions for future work. For example, how could this work inform improved aerosol modelling, observational networks, or policy-relevant assessments of air quality and climate?'*

**Response:** The final paragraph was rewritten to include implications, limitations, and directions for future work as follows:

Beyond the descriptive patterns described above, the results have several implications for how aerosol variability should be represented and observed in this region. The frequent occurrence of mixed aerosol states, in particular the recurring continental-smoke and dust-smoke combinations identified by NATALI, points to a need for regional chemical transport models to better resolve vertically structured mixing and the seasonally varying ageing of fine-mode particles. The extended record also demonstrates the importance of maintaining ACTRIS-grade, co-located lidar and photometer observations at sites where multiple transport pathways intersect, since column-only products struggle to capture the layered structure revealed here. At the same time, the analysis is constrained by reduced lidar sampling during cloudy and winter periods, the limited capability of the low-resolution NATALI mode to separate fine-mode types with similar spectral behaviour, and the relatively small set of layers with calibrated depolarisation for the high-resolution classification. Addressing these limitations will require expanding the multi-parameter retrieval capability toward combined lidar-radiometer inversions and integrating these with regional modelling tools such as FLEXPART-CTM coupling. Such developments would allow a more quantitative assessment of source contributions and radiative effects, and would strengthen the use of long-term ACTRIS datasets for evaluating emission-reduction policies and constraining aerosol-climate interactions in Eastern Europe.
* * *
**Specific comments**

*"• Revise your citations. When citing, correct is 'Nicolae et al. (2023) have found …' not '(Nicolae et al., 2023) have found…'. E.g. lines 130, 165, 213 etc Improve the abstract."*

**Response:** Citations have been corrected and the abstract has been improved.
* * *
*"Improve all figures."*

**Response:** All figures have been improved as requested.
* * *
*"Line 93: 'from the'"*

**Response:** Corrected.
* * *
*"Line 228: cite the OPAC database"*

**Response:** Citation added.

Hess, M., P. Koepke, and I. Schult, 1998: Optical Properties of Aerosols and Clouds: The Software Package OPAC. Bull. Amer. Meteor. Soc., 79, 831–844, https://doi.org/10.1175/1520-0477(1998)079<0831:OPOAAC>2.0.CO;2.
* * *
*"Line 244: review the aerosol type classification."*

**Response:** The classification scheme as is described in details in Nicolae et al,2018) has been fully reviewed and written also in this paper as follows:

"The main aerosol category is determined across six possible outputs, each representing a dominant type that may include up to 50 percent presence of other components as minor traces: continental, continental polluted, smoke, dust, marine, and mineral mixtures. In high-resolution mode, NATALI further refines these classes into a set of fourteen advanced aerosol types that capture both pure and mixed states. These include: Continental (water soluble, insoluble, soot), Dust (mineral nucleation, mineral accumulation, mineral coarse mode, water soluble, soot), Continental Polluted (water soluble, soot, insoluble, sulfate), Marine (water soluble, sea salt accumulation mode, sea salt coarse mode, soot), Smoke (water soluble, soot, sulfate), Volcanic (mineral nucleation, mineral accumulation, mineral coarse mode, water soluble, sulfate), as well as mixed and binary types such as ContinentalDust (continental + dust), MarineMineral (dust + marine), ContinentalSmoke (continental + smoke), DustPolluted (dust + smoke or dust + continental polluted), Coastal (continental + marine), CoastalPolluted (continental polluted + marine), MixedDust (continental + dust + marine), and MixedSmoke (continental + smoke + marine). The correspondence between the aerosol types retrieved by NATALI in high- and low-resolution modes is presented in the results section."

References:

Adam, M., Nicolae, D., Stachlewska, I. S., Papayannis, A., and Balis, D.: Biomass burning events measured by lidars in EARLINET – Part 1: Data analysis methodology, Atmos. Chem. Phys., 20, 13905–13927, https://doi.org/10.5194/acp-20-13905-2020, 2020.

Ansmann, A., Ohneiser, K., Mamouri, R.-E., Knopf, D. A., Veselovskii, I., Baars, H., Engelmann, R., Foth, A., Jimenez, C., Seifert, P., and Barja, B.: Tropospheric and stratospheric wildfire smoke profiling with lidar: mass, surface area, CCN, and INP retrieval, Atmos. Chem. Phys., 21, 9779–9807, https://doi.org/10.5194/acp-21-9779-2021, 2021

del Águila, A., Ortiz-Amezcua, P., Tabik, S., Bravo-Aranda, J. A., Fernández-Carvelo, S., and Alados-Arboledas, L.: Aerosol type classification with machine learning techniques applied to multiwavelength lidar data from EARLINET, Atmos. Chem. Phys., 25, 12549–12567, https://doi.org/10.5194/acp-25-12549-2025, 2025.

Floutsi, A. A., Baars, H., Engelmann, R., Althausen, D., Ansmann, A., Bohlmann, S., Heese, B., Hofer, J., Kanitz, T., Haarig, M., Ohneiser, K., Radenz, M., Seifert, P., Skupin, A., Yin, Z., Abdullaev, S. F., Komppula, M., Filioglou, M., Giannakaki, E., Stachlewska, I. S., Janicka, L., Bortoli, D., Marinou, E., Amiridis, V., Gialitaki, A., Mamouri, R.-E., Barja, B., and Wandinger, U.: DeLiAn – a growing collection of depolarization ratio, lidar ratio and Ångström exponent for different aerosol types and mixtures from ground-based lidar observations, Atmos. Meas. Tech., 16, 2353–2379, https://doi.org/10.5194/amt-16-2353-2023, 2023.

Hess, M., P. Koepke, and I. Schult, 1998: Optical Properties of Aerosols and Clouds: The Software Package OPAC. Bull. Amer. Meteor. Soc., 79, 831–844, https://doi.org/10.1175/1520-0477(1998)079<0831:OPOAAC>2.0.CO;2.

Lucja Janicka, Lina Davuliene, Steigvile Bycenkiene, and Iwona S. Stachlewska, "Long term observations of biomass burning aerosol over Warsaw by means of multiwavelength lidar," Opt. Express 31, 33150-33174 (2023)

Nicolae, D., Nemuc, A., Müller, D., Talianu, C., Vasilescu, J., Belegante, L., and Kolgotin, A.: Characterization of fresh and aged biomass burning events using multiwavelength Raman lidar and mass spectrometry, J. Geophys. Res.-Atmos., 118, 2956–2965, https://doi.org/10.1002/jgrd.50324, 2013.

Mylonaki, M., Giannakaki, E., Papayannis, A., Papanikolaou, C.-A., Komppula, M., Nicolae, D., Papagiannopoulos, N., Amodeo, A., Baars, H., and Soupiona, O.: Aerosol type classification analysis using EARLINET multiwavelength and depolarization lidar observations, Atmos. Chem. Phys., 21, 2211–2227, https://doi.org/10.5194/acp-21-2211-2021, 2021.

Papagiannopoulos, N., Mona, L., Amodeo, A., D'Amico, G., Comeron, A., Rodriguez-Gomez, A., Sicard, M. An automatic observation-based aerosol typing method for EARLINET. "Atmospheric chemistry and physics", 6 Novembre 2018, vol. 18, núm. 21, p. 15879-15901